# Gradient Scarcity in Graph Learning with Bilevel Optimization

**Hashem Ghanem**  *hashemghanem234@gmail.com*
*Institut de Mathématiques de Bourgogne, CNRS, Université de Bourgogne, France*

**Samuel Vaiter**  *samuel.vaiter@cnrs.fr*
*CNRS & Université Côte d'Azur, Laboratoire J. A. Dieudonné*

**Nicolas Keriven**  *nicolas.keriven@cnrs.fr*
*CNRS & IRISA - Institut de Recherche en Informatique et Systèmes Aléatoires*

Reviewed on OpenReview: *https://openreview.net/forum?id=10YJTIsVYq*

## Abstract

*Gradient scarcity* emerges when learning graphs by minimizing a loss on a subset of nodes under the semi-supervised setting. It consists in edges between unlabeled nodes that are far from the labeled ones receiving zero gradients. The phenomenon was first described when jointly optimizing the graph and the parameters of a shallow Graph Neural Network (GNN) using a single loss function. In this work, we give a precise mathematical characterization of this phenomenon, and prove that it also emerges in *bilevel* optimization. While for GNNs gradient scarcity occurs due to their finite receptive field, we show that it also occurs with the Laplacian regularization as gradients decrease exponentially in amplitude with distance to labeled nodes, despite the infinite receptive field of this model. We study several solutions to this issue including latent graph learning using a Graph-to-Graph model (G2G), graph regularization to impose a prior structure on the graph, and reducing the graph diameter by optimizing for a larger set of edges. Our empirical results validate our analysis and show that this issue also occurs with the Approximate Personalized Propagation of Neural Predictions (APPNP), which approximates a model of infinite receptive field.

## 1 Introduction

Semi-supervised learning is learning from datasets comprising both labeled and unlabeled points. This problem is usually handled with extra assumptions on the data. The main one, called *homophily*, refers to the fact that "nearby" points are likely to have similar labels (Wang & Zhang, 2006). Moreover, points in many applications represent entities that are naturally linked to each other, *e.g.,* in biology (Liu et al., 2018) or social media (Liben-Nowell & Kleinberg, 2003). There again, linked entities are likely to share the same label, which underlines the importance of exploiting the links when solving semi-supervised learning problems. Consequently, various graph-based methods have been developed for semi-supervised learning.

One issue with such methods is that their performance is highly dependent on the graph quality. This issue poses a significant challenge as real-world graphs are inherently noisy, significantly degrading the performance and leading to sub-optimal solutions. Many *graph learning* algorithms have thus been proposed in the literature to overcome this issue. Among these methods, a mainstream approach is to optimize the graph structure by means of optimizing the performance in the downstream task.

This approach involves generating a graph that, when used by a graph-based method, minimizes some labeling loss function. However, graph-based methods themselves require an optimization process to minimize a labeling loss function too. This problem can be formulated in two different manners. In the *joint* optimization formulation, a single loss function is used to assess both the graph and the graph-based model,

and both objects are optimized simultaneously. In the *bilevel* optimization formulation, each of the graph and the graph-based model is assessed by its own loss function. In this scenario, the graph loss function takes in input the optimized graph-based model trained while fixing the graph, *i.e.*, we have a constrained optimization problem where the constraint involves the output of another optimization problem. In both cases, the optimization is usually handled using gradient-based methods.

For shallow Graph Neural Networks (GNNs), Fatemi et al. (2021) show that joint optimization leads to *gradient scarcity*. It refers to the fact that connections between unlabeled nodes "far" from the labeled ones receive *zero gradients*, *i.e.*, they receive no supervision during the optimization and are not learned. This is due to the finite receptive field of message-passing GNNs, which is defined as the neighborhood range captured by a node during the message passing phase. In this work, we focus on bilevel optimization and prove that gradient scarcity also occurs for shallow GNNs, despite the additional dependency between the parameters in this setting. We also show that this issue emerges with other graph-based models. We do that theoretically and empirically for the Laplacian-based label propagation, which, unlike GNNs, has an infinite receptive field, and empirically for the Approximate Personalized Propagation of Neural Predictions (APPNP) (Gasteiger et al., 2018), which approximates a model of infinite receptive field.

The rest of the paper is organized as follows. The rest of this section presents the considered graph-based models for semi-supervised learning, and the bilevel optimization framework. Section 2 reviews related work. Sections 3 and 4 presents our theoretical analysis of gradient scarcity for shallow GNNs and the Laplacian regularization model, respectively. Section 5 proposes three strategies that can be used to mitigate this issue. We finally present our empirical results in Section 6.

## 1.1 Graph-based models for semi-supervised learning

A graph $\mathcal{G}$ is a pair $(V, E)$, where $V$ is a set of $n$ nodes and $E \subseteq V \times V$ is a set of edges. We represent a graph by its adjacency matrix $\boldsymbol{A} \in \mathbb{R}^{n \times n}$, where $\boldsymbol{A}_{i,j}$ is the weight of the edge between nodes $i, j$. We denote by $\boldsymbol{X} \in \mathbb{R}^{n \times p}$ the feature matrix whose rows include the features of corresponding nodes.

**Definition 1.1** (Distance between two nodes). We say that node $i$ is $k$-hop from node $j$ if the minimum number of edges forming a path from $i$ to $j$ is $k$.

**Definition 1.2** (Node distance to a subset of nodes). A node $i$ is said to be $k$-hop from a subset of nodes if the minimum number of edges forming a path from $i$ to a node in the subset is $k$. We further refer to the set of nodes that are at most $k$-hop from $i$ by its $k$-hop neighborhood.

**Definition 1.3** (Edge distance to a subset of nodes). The distance of an edge $(i, j)$ to a subset of nodes equals the minimum between the distance of $i$ and the distance of $j$ to this subset.

We look at transductive semi-supervised learning problems, where we have a set of points, a subset of which is labeled, and the goal is to approximate the labeling function on unlabeled points. Formally, we have $(\boldsymbol{X}_{obs}, \mathcal{G}_{obs}, \boldsymbol{Y}_{obs})$, where $\mathcal{G}_{obs}$ is the observed graph, $\boldsymbol{X}_{obs}$ are the observed node features (we will drop the subscript and write $\boldsymbol{X}$ in the rest of the paper) and $\boldsymbol{Y}_{obs} \in \mathbb{R}^n$ contains the labels of a subset of points $V_{tr}$, *i.e.*, it containes the labels at coordinates $i \in V_{tr} \subset V$ and, *e.g.*, not-a-number "$NaN$" outside of $V_{tr}$. There are roughly two main strategies to solve semi-supervised learning problems. The first is to *propagate* known labels using a *regularization* process. Predicted labels read the following:

$$\boldsymbol{Y}_{\mathrm{Reg}}(\boldsymbol{A}) \in \operatorname*{arg\,min}_{\boldsymbol{Y} \in \mathcal{B}} \frac{1}{|V_{tr}|} \sum_{i \in V_{tr}} \ell(\boldsymbol{Y}_i, (\boldsymbol{Y}_{obs})_i) + \lambda R(\boldsymbol{Y}, \boldsymbol{A}), \tag{1}$$

where $\mathcal{B}$ is an admissible set, $\ell$ is usually a smooth loss function, $R$ is a regularization term, and $\lambda$ is a balancing parameter. In regression tasks, $\mathcal{B}$ is commonly the space $\mathbb{R}^n$, and $\ell$ is chosen to be the Mean Squared Error (MSE). Whilst in classification tasks, the $i$-th element $\boldsymbol{Y}_i$ is not a scalar but rather a vector holding the probability distribution over classes. Formally, $\mathcal{B} = \{\boldsymbol{Y} \in \mathbb{R}^{n \times C} \mid \forall i, \sum_{c=1}^{C} \boldsymbol{Y}_{i,c} = 1, \forall i, c, \boldsymbol{Y}_{i,c} \geq 0\}$, where $C$ is the number of classes. In this case, $\ell$ is the Categorical Cross Entropy (CCE) loss function.

In fact, graph regularization methods usually differ from each other by the choice of the regularization function $R$. A popular choice is the *Laplacian regularization*, (Slepcev & Thorpe, 2019; Pang & Cheung,

2017):

$$R(\boldsymbol{Y}, \boldsymbol{A}) = \frac{1}{|E|} \sum_{i,j} \boldsymbol{A}_{i,j} \|\boldsymbol{Y}_i - \boldsymbol{Y}_j\|_2^2 \ . \tag{2}$$

Note that here the node features $\boldsymbol{X}$ are not used.

The second main strategy in semi-supervised learning is to train a *parametric model* $\boldsymbol{Y}_W(\boldsymbol{X}, \boldsymbol{A})$ parameterized by the parameters $W$, such as GNNs. The objective reads:

$$\boldsymbol{Y}_{GNN}(\boldsymbol{A}) = \boldsymbol{Y}_{W^\star}(\boldsymbol{X}, \boldsymbol{A}), \text{where}$$
$$W^\star = \arg\min_W \frac{1}{|V_{tr}|} \sum_{i \in V_{tr}} \ell\Big( \big(\boldsymbol{Y}_W(\boldsymbol{X}, \boldsymbol{A})\big)_i, (\boldsymbol{Y}_{obs})_i \Big) \ . \tag{3}$$

In this paper, we use message-passing GNNs with sum aggregation (Morris et al., 2019). The first layer is $\boldsymbol{X}^{[0]} = \boldsymbol{X}$, propagated as

$$\boldsymbol{X}^{[l]} = \phi(\boldsymbol{X}^{[l-1]} \boldsymbol{W}_1^{[l]} + \boldsymbol{A} \boldsymbol{X}^{[l-1]} \boldsymbol{W}_2^{[l]} + \mathbf{1}_n (\boldsymbol{b}^{[l]})^\top) \ , \tag{4}$$

where $\boldsymbol{W}_1^{[l]}, \boldsymbol{W}_2^{[l]} \in \mathbb{R}^{d_{l-1} \times d_l}$ are learnable parameters, $\boldsymbol{b}^{[l]} \in \mathbb{R}^{d_l}$ is a learnable bias, $d_l$ is the output dimensionality of the $l$-th layer, $\mathbf{1}_n = [1, \ldots, 1]^\top \in \mathbb{R}^n$, and $\phi$ is a non-linear function applied element-wise. The output $\boldsymbol{Y}_W(\boldsymbol{X}, \boldsymbol{A}) = \boldsymbol{X}^{[k]}$ is obtained after $k$ rounds of message passing, and the parameters are gathered as $W = \{\boldsymbol{W}_1^{[l]}, \boldsymbol{W}_2^{[l]}, \boldsymbol{b}^{[l]}\}_{l=1}^k$.

Another advanced example of parametric models is the Personalized Propagation of Neural Predictions (PPNP) and its fast approximation APPNP (Gasteiger et al., 2018), which enable an infinite receptive field unlike shallow GNNs such as the model in Eq. (4). Note that the Laplacian regularization model enjoys an infinite receptive field as well. In the presented material, we analyze the gradient scarcity phenomenon for both GNNs and the Laplacian regularization, which is possible thanks to their simple models, and we empirically verify that it occurs with APPNP.

## 1.2 Joint and bilevel optimization for graph learning

We learn a graph by learning its adjacency matrix. In the joint optimization setting, the graph and the graph-based model are optimized simultaneously by minimizing a single loss function, usually using a gradient-based algorithm. For instance, denoting by $W$ the parameters a GNN model, the joint optimization reads:

$$\min_{\boldsymbol{A}, W} F = \frac{1}{|V_{tr}|} \sum_{i \in V_{tr}} \ell\Big( \big(\boldsymbol{Y}_W(\boldsymbol{X}, \boldsymbol{A})\big)_i, (\boldsymbol{Y}_{obs})_i \Big) \ , \tag{5}$$

In this scenario, $\boldsymbol{A}$ and $W$ are simultaneously updated in each iteration of the gradient-based algorithm.

In the bilevel optimization scenario, the graph has its own labeling loss function, which is a function of the *optimized* graph-based model. In the theoretical part of this paper, we analyze gradient scarcity under the bilevel optimization setting when the graph-based model is either a GNN or the Laplacian regularization. Therefore, we restrict the following formulation to these two cases. Given a second set of labeled nodes $V_{out} \subset V$ distinct from $V_{tr}$, and a set of admissible adjacency matrices $\mathcal{A}$, the bilevel optimization is cast as

$$\min_{\boldsymbol{A} \in \mathcal{A}} F_{out}(\boldsymbol{A}) = \frac{1}{|V_{out}|} \sum_{i \in V_{out}} \ell(\boldsymbol{Y}(\boldsymbol{A})_i, (\boldsymbol{Y}_{obs})_i), \tag{6a}$$

$$\text{s.t.} \quad \boldsymbol{Y}(\boldsymbol{A}) = \boldsymbol{Y}_{GNN}(\boldsymbol{A}) \quad \text{or} \quad \boldsymbol{Y}(\boldsymbol{A}) = \boldsymbol{Y}_{\text{Reg}}(\boldsymbol{A}) \ . \tag{6b}$$

That is, the minimization of the objective function $F_{out}$, called the *outer* optimization problem, involves $\boldsymbol{Y}(\boldsymbol{A})$, which is itself the result of an *inner* optimization problem, either equation 3 over $W$ or equation 1 over $\boldsymbol{Y}$. The objective function $F_{out}$ is used to optimize the graph, and the objective function $F_{in}$ is used to optimize the graph-based model. Several models are possible for $\mathcal{A}$:

**Full learning**: $\mathcal{A} = [a,b]^{n \times n}$ is the set of all weighted adjacency matrices (generally with some bounds $a, b$ on the weights). This choice necessarily leads to an impractical quadratic complexity on the minimization.

**Edge refinement**: the learned adjacency matrix has the same zero-pattern as the observed adjacency matrix, that is, we learn weights only on existing edges.

$$\mathcal{A} = \{\boldsymbol{A} \in [a,b]^{n \times n} | \boldsymbol{A}_{ij} = 0 \text{ when } (\boldsymbol{A}_{obs})_{ij} = 0\}.$$

The complexity is proportional to the number of edges, generally less than quadratic in $n$ as graphs tend to be sparse, which makes edge refinement more scalable than the full learning setting.

**Generalized edge refinement**: same principle, but the zero-pattern is given by a modification of the observed adjacency matrix. For instance, taking the zero-pattern of $\boldsymbol{A}_{obs}^r$ yields an edge between nodes that are less than $r$-hop from each other in $\mathcal{G}_{obs}$, where nodes $i$ and $j$ are $r$-hop from each other if the length of the shortest path between them is $r$.

**Latent graph learning**: the learned graph is the output of a parametric model, that takes as input the node features and the observed graph: $\mathcal{A} = \{\boldsymbol{A} = f_\theta(\boldsymbol{A}_{obs}, \boldsymbol{X})\}$. We will refer to such models as *Graph-to-Graph* (G2G).

Both the inner and the outer problems are treated by gradient-based algorithms. We refer to the outer gradient $\nabla F_{out}$, whether with respect to $\boldsymbol{A}$ or to $\theta$, as *hypergradient*.

To our knowledge, the literature has yet to present benchmarks comparing the bilevel and joint frameworks in graph learning. However, we can distinguish two primary differences between them. Firstly, the bilevel framework treats the graph as a hyper-parameter, while the joint framework perceives it in a similar way to the GNN parameters. Secondly, the joint framework is more prone to overfitting. This issue is commonly addressed through graph regularization and similar techniques (Wang & Leskovec, 2020; Fatemi et al., 2021). Exploring further comparisons between these frameworks, particularly in terms of their performance and limitations, is an intriguing direction for our future research.

## 1.3 The resolution of the joint and the bilevel optimization problems

In fact, the solution of any of the two problems cannot be exactly computed. For the joint optimization problem 5, the solution does not enjoy a closed-form expression that can be evaluated. In general, this problem is addressed using gradient-based algorithms, where both $\boldsymbol{A}, W$ are updated at every iteration in order to converge to a "good" local minimum.

For the bilevel optimization problem 6, the problem is intractable since neither the solution of the inner problem nor its gradient *w.r.t.* $\boldsymbol{A}$ (or to $\theta$ with G2G models) has a closed-form expression that can be evaluated. To overcome this difficulty, we unroll (Gregor & LeCun, 2010) $\tau_{in}$ iterations of a gradient-based optimizer applied on the inner problem, then use higher-order Automatic Differentiation (AD) to trace these iterations and approximate the hypergradient. In this work, we use the `Higher` package (Grefenstette et al., 2019) to perform the aforementioned automatic differentiation.

## 1.4 Contributions

Previous works observed gradient scarcity when optimizing the graph and a *shallow* GNN using joint optimization. Indeed, a $k$-layer GNN computes the label of a node using information from $r$-hops far nodes with $r \leq k$. This label is then not a function of edges connecting nodes outside this neighborhood, and the term in the labeling loss corresponding to this label returns null gradients on those distant edges. However, it is not straightforward how to extend this argument for bilevel optimization. Specifically, the previous discussion necessitates that the derivatives of the GNN parameters *w.r.t.* the distant edges equal zero after every gradient-based update. This applies to the joint optimization setting by construction, but is not easy to prove in the bilevel optimization setting where additional dependency exists between the parameters of the problem. Moreover, if the problem holds in the bilevel optimization setting, the roles of $V_{tr}$ and $V_{out}$ need to be clarified. Another question is if this problem is mitigated by resorting to graph-based models with infinite receptive field, *e.g.,* the Laplacian regularization and APPNP.

In this work, **we prove that hypergradient scarcity occurs under the bilevel optimization setting when adopting shallow GNNs as the graph-based model.** We show that using a $k$-layer GNN induces null hypergradients on edges between nodes at least $k$-hop from labeled nodes in $V_{tr} \cup V_{out}$. **For the Laplacian regularization, we prove that the problem persists**, as hypergradients are exponentially damped with distance from labeled nodes. **We empirically validate our findings and show that the problem occurs with the APPNP model as well**. Then, we test three possible strategies to solve this issue: refining a power of the given adjacency matrix, graph regularization, and latent graph learning with G2G models. Furthermore, we empirically **distinguish between hypergradient scarcity and overfitting**, in the sense that solving the former does not necessarily resolve the latter. To the best of our knowledge, this is the first work that mathematically tackles the gradient scarcity problem in graph learning with bilevel optimization, and examines the phenomenon for models with infinite receptive field.

## 2 Related work

**Bilevel optimization** has many applications like multi-task and meta learning (Bennett et al., 2006; Flamary et al., 2014; Franceschi et al., 2018). See Colson et al. (2007) for a comprehensive review of applications.

**Graph neural networks GNNs.** Different GNNs implement different flavors of message passing. One of the simplest GNNs is the model in Eq. (4) (Morris et al., 2019). Another influential version is the GCN model (Kipf & Welling, 2017), which simplifies the spectral convolution on graphs to a first-order approximation. Other popular variants include $S^2GC$ (Zhu & Koniusz, 2020), GNN-LF/HF (Zhu et al., 2021), graph autoencoders (GAEs) (Kipf & Welling, 2016), graph generative adversarial networks (Liu et al., 2019), etc. These models address different challenges, such as scalability, interpretability, and robustness.

A limitation of GNNs is their finite, usually small, receptive field, which cannot be increased by adding more layers due to the oversmoothing problem (Keriven, 2022). One line of work to address this issue includes deep GNNs, which can leverage a higher number of layers (Xu et al., 2018; Huang et al., 2018; Li et al., 2019; He et al., 2016a;b; Ruiz et al., 2020; Zhang et al., 2022). A second line of work focuses on the way graph information is exploited rather than on the model structure (Wu et al., 2019; Liu et al., 2020; Rossi et al., 2020; Li et al., 2020). For instance, PPNP, its approximation APPNP (Gasteiger et al., 2018) and PPRGo (Bojchevski et al., 2020) use the personalized PageRank (Page, 1998) method to propagate node information to the entire graph while avoiding the oversmoothing problem. PPNP and APPNP have produced state-of-the-art results on several node classification datasets. We later theoretically show that the GNN in Eq. (4) promotes the hypergradient scarcity problem when adopted in the bilevel framework. We empirically show the same for APPNP, even though it approximates an infinite receptive field.

**Graph learning.** One simple learner is the $k$-Nearest Neighbors ($k$-NN) technique and its variants (Roweis & Saul, 2000; Tenenbaum et al., 2000; Zhu et al., 2003), where a similarity metric is deployed to detect the $k$ most similar nodes to each node. Another line of research focused on assigning weights to observed edges or to edges constructed with $k$-NN-like methods using a similarity metric (Gretton et al., 2006; Zhu et al., 2003; Kapoor et al., 2005; Li et al., 2018). Locality-inducing methods optimize edge weights in a given graph by promoting the assumption that each node can be produced by an edge-weighted sum of its neighbors (Saul & Roweis, 2003; Wang & Zhang, 2006; Daitch et al., 2009). This methodology is usually noise sensitive and prone to overfitting as the number of edges is larger than the number of nodes (Qiao et al., 2018). Smoothness-inducing methods learn a graph that promotes smoothness seen in node features (Hu et al., 2013; Kalofolias, 2016). This framework costs $\mathcal{O}(n^2)$ so it does not scale well . However, some approximations reduced the complexity to $\mathcal{O}(n \log(n))$ (Kalofolias & Perraudin, 2017).

Another mainstream approach is to optimize the graph such that it improves the labelling performance in the downstream task. Such works usually optimize together the graph and the graph-based model. Stretcu et al. (2019) proposed the deep model GAM, which is trained by penalizing the absence of an edge between nodes with the same label. Wang & Leskovec (2020) optimize the edge weights and a GNN model using joint optimization. To alleviate overfitting resulting from learning a large number of parameters, the authors regularize the graph using the label propagation model (Zhu, 2005). Similarly, Fatemi et al. (2021) use another technique to regularize the graph. Attention mechanisms evaluate edge weights after every GNN layer based on node similarity (Luong et al., 2015; Vaswani et al., 2017; Veličković et al., 2018; Kim & Oh, 2021).

The previous methods produced state-of-the-art results on many node classification tasks. Here, we consider the graph learning problem when formulated as a *supervised* bilevel optimization problem. Franceschi et al. (2019) adopted bilevel optimization to learn the parameters of Bernoulli probability distributions over independent random edges. This method produced competitive results but it includes learning $n^2$ parameters which limits scalability. Wan & Kokel (2021) use the bilevel framework to sparsify observed graphs while keeping it connected. Resulted graphs do not necessarily outperform the observed ones.

**Gradient scarcity** was identified by Fatemi et al. (2021), where the authors optimized the graph and a shallow GNN model using joint optimization. The authors showed that when adopting a $k$-layer GNN classifier (with $k = 2$ in their case), edges between unlabeled nodes do not receive any supervision if they are at least 2-hop from labeled nodes. They referred to this issue as the *supervision starvation problem.* They quantify the starvation for the special case of Erdös-Rényi graphs. Note that gradient scarcity and supervision starvation refer to the same phenomenon. This issue cannot be resolved by adding more layers to the GNN as more data and labels will be needed for training, and due to the oversmoothing problem. To overcome this issue, authors proposed to regularize the graph by enforcing the assumption that a good graph must also perform well in denoising node features.

That said, authors implicitly assumed no dependence between the GNN parameters and the graph when identifying gradient scarcity, which is the case in joint optimization but not in bilevel optimization. To the best of our knowledge, this issue has not yet been studied for the bilevel optimization setting. Moreover, it is not clear if this problem is resolved with graph-based models with infinite receptive field, *e.g.,* the Laplacian regularization and APPNP. We treat both these topics in our work.

Liu et al. (2022) stated that optimizing both the graph and a GNN model under the supervision of a classification task introduces reliance on available labels, bias in the edge distribution and even reduces the span of potential application tasks. Still, this statement is not accompanied with a theoretical justification. To overcome this problem, authors suggested to avoid label-based graph optimization, and proposed an *unsupervised* graph learning framework. Although the unsupervised framework proved effective and competed state-of-the-art methods, we believe that labels contain informative knowledge that is not exploited when deploying unsupervised learners, and that better results are obtained by getting the best of both worlds.

## 3  Hypergradient scarcity with shallow GNNs

In this section, we consider the bilevel optimization 6 with the GNN model in Eq. (4), *i.e.,* $\boldsymbol{Y}(\boldsymbol{A}) = \boldsymbol{Y}_{GNN}(\boldsymbol{A}) = \boldsymbol{Y}_{W^\star}(\boldsymbol{A}, \boldsymbol{X})$. We place ourselves under the *edge refinement* setting where we optimize the weight of every existing edge in $\boldsymbol{A}_{obs}$.

In Fatemi et al. (2021), the authors demonstrated that the predicted node label using a 2-layer GNN integrates information from nodes of distance less than two hops, *i.e.,* the label is not a function of edges connecting nodes at least 2-hop far away. Consequently, when optimizing the graph by minimizing the classification error of that label via a gradient-based algorithm, these edges receive zero-valued gradients. However, the authors used joint optimization where there is no dependency between $W$ and $\boldsymbol{A}$, *i.e.,* $\boldsymbol{J}_W(\boldsymbol{A}) = \boldsymbol{0}$ at every gradient-based iteration. *This is not the case for bilevel optimization.* In this section, we first examine the joint optimization scheme, and prove the existence of the problem for a generic number of layers $k$, similar to Fatemi et al. (2021). For the bilevel optimization setting, we then prove that the optimized parameters $W^\star$ are not a function of edges connecting nodes at least $k$-hop from nodes in $V_{tr}$. After that, we conclude that hypergradient scarcity holds in the bilevel setting for edges connecting nodes at least $k$-hop from nodes in the *union* $V_{tr} \cup V_{out}$.

### 3.1  Gradient scarcity for joint optimization

In this initial result, we assume that the parameters $W$ do not depend on $\boldsymbol{A}$, *i.e.,* $W$ is not a function of $\boldsymbol{A}$. Indeed, by construction, this is the case after every gradient-based iteration $t$ in the joint optimization setting, where we have $\frac{\partial W_t}{\partial \boldsymbol{A}_t} = \boldsymbol{0}$, with $W_t, \boldsymbol{A}_t$ being the updated states of $W, \boldsymbol{A}$ at iteration $t$, respectively. This hypothesis is not satisfied in the bilevel optimization setting, which we will address in subsequent sections. The next theorem shows gradient scarcity in the joint optimization setting by examining $\boldsymbol{Y}_W(\boldsymbol{A}, \boldsymbol{X})$.

**Theorem 3.1.** *Let $\boldsymbol{Y}_W = \boldsymbol{Y}_W(\boldsymbol{A}, \boldsymbol{X})$ be the output of a $k$-layer GNN parameterized by $W$ as in Eq. (4). Let $i, j, u$ be such that nodes $i, j$ are at least $k$-hop from node $u$. Assume that $\frac{\partial W}{\partial \boldsymbol{A}_{i,j}} = \boldsymbol{0}$. Then we have*

$$\frac{\partial (\boldsymbol{Y}_W)_u}{\partial \boldsymbol{A}_{i,j}} = 0 \quad . \tag{7}$$

*Proof.* The proof is done by induction on $k$. For $k = 1$, this is indeed the case since $\boldsymbol{X}^{[0]} = \boldsymbol{X}$ does not depend on $\boldsymbol{A}$, and that $\boldsymbol{A}_{i,j}$ does not belong to the row $\boldsymbol{A}_{u,:}$ which is the only row in $\boldsymbol{A}$ that contributes in the value $(\boldsymbol{X}^{[1]})_{u,:}$.

Assume that the statement is true for some arbitrary positive integer $k$, we show that it is also true for $(k+1)$. If two nodes $i, j$ are at least $(k+1)$-hop far from node $u$, then clearly they are at least $k$-hop far from it too. Thus from the induction assumption, we have that $(\boldsymbol{X}^{[k]})_{u,:}$ is independent of $\boldsymbol{A}_{i,j}$. Also, $\boldsymbol{W}_1^{[k+1]}$ does not depend on $\boldsymbol{A}_{i,j}$ since we assume $\frac{\partial W}{\partial \boldsymbol{A}_{i,j}} = \boldsymbol{0}$. Therefore, $(\boldsymbol{X}^{[k]} \boldsymbol{W}_1^{[k+1]})_{u,:}$ in Eq. (4) does not depend on $\boldsymbol{A}_{i,j}$ too.

In a similar way, if $i, j$ are at least $(k+1)$-hop far from $u$, then they are at least $k$-hop far from any of its neighbors $v$ where $\boldsymbol{A}_{u,v} \neq 0$. Therefore, for all $v$, $\boldsymbol{A}_{u,v} \neq 0$, then $\frac{\partial (\boldsymbol{X}^{[k]})_{v,:}}{\partial \boldsymbol{A}_{i,j}} = \boldsymbol{0}$. Moreover, $\frac{\partial \boldsymbol{W}_2^{[k+1]}}{\partial \boldsymbol{A}_{i,j}} = \boldsymbol{0}$ since we assume $\frac{\partial W}{\partial \boldsymbol{A}_{i,j}} = \boldsymbol{0}$. This makes $(\boldsymbol{A}\boldsymbol{X}^{[k]} \boldsymbol{W}_2^{[k+1]})_{u,:} = \boldsymbol{A}_{u,:} \boldsymbol{X}^{[k]} \boldsymbol{W}_2^{[k+1]}$ in Eq. (4) independent of $\boldsymbol{A}_{i,j}$. This concludes the proof, as $\frac{\partial (\boldsymbol{Y}_W)_u}{\partial \boldsymbol{A}_{i,j}} = \frac{\partial (\boldsymbol{X}^{[k+1]})_u}{\partial \boldsymbol{A}_{i,j}} = \boldsymbol{0}$. □

Note that the number of hops $k$ that defines edges affected by gradient scarcity equals the number of the GNN layers $k$. That is, Theorem 3.1 shows that the labels $(\boldsymbol{Y}_W)_u$ on training nodes $u \in V_{tr}$ are not a function of edges between nodes at least $k$-hop from these training nodes. Consequently, the gradient of the labeling loss function in Eq. (5) equals zero on these edges.

## 3.2 The gradient of the optimized parameters

With the assumption that $W$ is not a function of the adjacency matrix $\boldsymbol{A}$, Theorem 3.1 states that edges between nodes at least $k$-hop from the training nodes receive no supervision. However, in the bilevel optimization scenario, after the first outer iteration $W$ may depend on $\boldsymbol{A}$. The next theorem shows that hypergradient scarcity still occurs in the bilevel optimization framework, as the "optimal" parameters used in practice are the result of a gradient-based algorithm. More precisely, we consider a sequence

$$W_{t+1} = W_t - Q_t(W_t, \nabla_{W_t} F_{in}) \quad , \tag{8}$$

where $Q_t$ is a smooth function. Note that $W_t$ does not necessarily converge towards the true optimal point $W^\star$.

**Theorem 3.2.** *Let $\boldsymbol{A}$ be an input graph to a $k$-layer GNN parameterized by $W$ as in Eq. (4), and $W_t$ be the output obtained by optimizing equation 3 for $W$ using a gradient-based iterates sequence. Let $i, j$ be nodes that are at least $k$-hop from any node in $V_{tr}$. Then, for all $t \in \mathbb{N}$,*

$$\frac{\partial W_t(\boldsymbol{A})}{\partial \boldsymbol{A}_{i,j}} = \boldsymbol{0} \quad . \tag{9}$$

*Proof.* The proof is carried out by induction on the iteration index $t$ of the gradient-based optimizer. Denote by $F_{in}$ the objective function in equation 3. For $t = 0$, $W_0$ is the initialization of $W$ which is usually random and does not depend on $\boldsymbol{A}$. For $t \geq 0$, we assume that $\frac{\partial W_t}{\partial \boldsymbol{A}_{i,j}} = \boldsymbol{0}$ and prove this must be true for $t + 1$. By differentiating Eq. (8) *w.r.t.* $\boldsymbol{A}_{i,j}$, and under the induction assumption, it is sufficient to show that $\frac{\partial Q_t(W_t, \nabla_{W_t} F_{in})}{\partial \boldsymbol{A}_{i,j}} = \boldsymbol{0}$ to prove that $\frac{\partial W_{t+1}(\boldsymbol{A})}{\partial \boldsymbol{A}_{i,j}} = \boldsymbol{0}$. Therefore, given the induction assumption and using the chain rule, proving that $\frac{\partial (\nabla_{W_t} F_{in})}{\partial \boldsymbol{A}_{i,j}} = \boldsymbol{0}$ is sufficient to complete the proof. The gradient $\nabla_{W_t} F_{in}$ writes:

$$\nabla_{W_t} F_{in} = \frac{1}{|V_{tr}|} \sum_{u \in V_{tr}} \nabla_{W_t} \ell\left( \left( \boldsymbol{Y}_{W_t}(\boldsymbol{X}, \boldsymbol{A}) \right)_u, (\boldsymbol{Y}_{obs})_u \right) \quad .$$

For all $u \in V_{tr}$, the term $\nabla_{W_t} \ell\left(\left(\boldsymbol{Y}_{W_t}(\boldsymbol{X}, \boldsymbol{A})\right)_u, (\boldsymbol{Y}_{obs})_u\right)$ might be a function of $\boldsymbol{A}_{i,j}$ through the terms $W_t$ and $\left(\boldsymbol{Y}_{W_t}(\boldsymbol{X}, \boldsymbol{A})\right)_u$. But $\frac{\partial W_t}{\partial \boldsymbol{A}_{i,j}} = \boldsymbol{0}$ from the induction assumption, and, given that, we have $\frac{\partial \left(\boldsymbol{Y}_{W_t}(\boldsymbol{X}, \boldsymbol{A})\right)_u}{\partial \boldsymbol{A}_{i,j}} = 0$ from Theorem 3.1. Thus, we have for all $u \in V_{tr}$, $\frac{\partial}{\partial \boldsymbol{A}_{i,j}} \nabla_{W_t} \ell\left(\left(\boldsymbol{Y}_{W_t}(\boldsymbol{X}, \boldsymbol{A})\right)_u, (\boldsymbol{Y}_{obs})_u\right) = \boldsymbol{0}$. This concludes the proof of Eq. (9) as it gives $\frac{\partial (\nabla_{W_t} F_{in})}{\partial \boldsymbol{A}_{i,j}} = \boldsymbol{0}$. $\qquad\square$

One notes that Theorem 3.2 still applies when the joint optimization problem 5 is augmented with a regularization term that regularizes the graph as in Wang & Leskovec (2020); Fatemi et al. (2021) for instance. For that this term does not affect updates performed on the parameters $W$. In this work, we are interested in studying the gradient scarcity phenomenon seen in the gradient of the labeling loss term as in Eq. (5). Therefore, we do not consider such regularization terms in our analysis. However, we will see in Sections 5 and 6 that graph regularization is one effective technique to alleviate gradient scarcity.

### 3.3 Hypergradient scarcity

Finally, we put the two previous results together. The next theorem states that within the bilevel optimization framework, edges between nodes at least $k$-hop from nodes in $V_{tr} \cup V_{out}$ receive zero hypergradients.

**Theorem 3.3.** *Let $\boldsymbol{Y}_W$ be a $k$-layer GNN following the model 4. Assume that the inner optimization problem is solved with a gradient-based algorithm as in Eq. (8). Then, for any pair of nodes $i, j$ at least $k$-hop from nodes in $V_{out} \cup V_{tr}$, we have $\frac{\partial F_{out}}{\partial \boldsymbol{A}_{i,j}} = \boldsymbol{0}$.*

*Proof.* Directly from Theorem 3.2 we have that $\frac{\partial W_t(\boldsymbol{A})}{\partial \boldsymbol{A}_{i,j}} = \boldsymbol{0}$ since $i, j$ are at least $k$-hop far from nodes in $V_{tr}$. This makes it possible to apply Theorem 3.1 to get that $\forall u \in V_{out}; \frac{\partial (\boldsymbol{Y}_{W_t})_u}{\partial \boldsymbol{A}_{i,j}} = \boldsymbol{0}$, as $i, j$ are at least $k$-hop far from nodes in $V_{out}$ and $\frac{\partial W_t(\boldsymbol{A})}{\partial \boldsymbol{A}_{i,j}} = \boldsymbol{0}$. This concludes the proof as $F_{out}$ penalizes the classification error only on nodes in $V_{out}$. $\qquad\square$

Theorem 3.3 shows that the hypergradient scarcity problem emerges when solving edge refinement tasks: if two nodes are at least $k$-hop far from nodes in $V_{out} \cup V_{tr}$ in $\boldsymbol{A}_{obs}$, the edge in between receives no hypergradients. In Section 5, we will examine several strategies to mitigate this phenomenon.

## 4 Hypergradient scarcity with the Laplacian regularization

In the previous section, we have seen how the finite receptive field of shallow GNNs directly induces the hypergradient scarcity problem. We now examine hypergradient scarcity when $\boldsymbol{Y}(\boldsymbol{A}) = \boldsymbol{Y}_{\mathrm{Reg}}(\boldsymbol{A})$ with the Laplacian regularization 2. Indeed, in this case the inner model does not have a finite receptive field, in the sense that in general $\frac{\partial \boldsymbol{Y}(\boldsymbol{A})}{\partial \boldsymbol{A}_{ij}} \neq 0$ for all $i, j$, unlike the shallow GNN case as proven by Theorem 3.1.

Surprisingly, we show that hypergradient scarcity still occurs in some sense. More precisely, we prove that the magnitude of hypergradients decreases exponentially with the sum of the distance to $V_{tr}$ and the distance to $V_{out}$.

To proceed, we consider regression downstream tasks as the inner minimizer $\boldsymbol{Y}(\boldsymbol{A})$ enjoys a closed-form expression in this case. Recall that in these tasks, we employ the MSE loss as $\ell$ in Eqs. (1) and (6). Let $\boldsymbol{S}_{in} \in \mathbb{R}^{n \times n}$ be the diagonal selection matrix whose entries equal 1 if the corresponding node is in $V_{tr}$ and 0 otherwise, the solution $\boldsymbol{Y}(\boldsymbol{A})$ reads:

$$\boldsymbol{Y}(\boldsymbol{A}) = \left(\tilde{\boldsymbol{S}}_{in} + \lambda \tilde{\boldsymbol{L}}\right)^{-1} \tilde{\boldsymbol{S}}_{in} \boldsymbol{Y}_{obs} \ ,$$

where $\tilde{\boldsymbol{S}}_{in} = \frac{\boldsymbol{S}_{in}}{|V_{tr}|}$, $\tilde{\boldsymbol{L}} = \frac{\boldsymbol{L}}{|E|}$, $\boldsymbol{L} = L(\boldsymbol{A}) = \boldsymbol{D} - \boldsymbol{A}$ is the Laplacian of the graph, and $\boldsymbol{D}$ is the diagonal degree matrix that includes node degrees on the diagonal: $\boldsymbol{D}_{i,i} = \sum_j \boldsymbol{A}_{i,j}$. For simplicity from now on, we denote $\boldsymbol{B} = \tilde{\boldsymbol{S}}_{in} + \lambda \tilde{\boldsymbol{L}}$. Then, we write $\boldsymbol{Y}(\boldsymbol{A})$ as:

$$\boldsymbol{Y}(\boldsymbol{A}) = \boldsymbol{B}^{-1} \tilde{\boldsymbol{S}}_{in} \boldsymbol{Y}_{obs} \ . \tag{10}$$

It is well-defined thanks to the following result.

**Lemma 4.1.** *Assume that the graph is connected. The eigenvalues $\mu_i$ of $\boldsymbol{B}$ satisfy, for all $i$:*

$$0 < \mu_{\min} \leq \mu_i \leq \mu_{\max} \leq \frac{1}{|V_{tr}|} + 2\lambda \ . \tag{11}$$

Given that, we now state the main result of this section.

**Theorem 4.2.** *Let nodes $i, j$ be at least $k$-hop from $V_{out}$, and $q$-hop from $V_{tr}$. Then we have:*

$$\left| \frac{\partial F_{out}}{\partial \boldsymbol{A}_{ij}} \right| \lesssim \lambda \frac{\sqrt{|V_{out}|} + \mu_{\min}\sqrt{|V_{tr}|}|V_{out}|}{\mu_{\min}^3 |V_{tr}||E|} y_\infty^2 (1 - \mu)^{q+k} \ , \tag{12}$$

*where $\mu = \frac{\mu_{\min}}{\mu_{\max}}$ and $y_\infty = \|\boldsymbol{Y}_{obs}\|_\infty$.*

Since both $\mu_{\min}, \mu_{\max}$ are strictly positive, as shown in the proof in Section 4.3, then $0 < 1 - \mu < 1$. Therefore, Theorem 4.2 states that the magnitude of the hypergradient is *exponentially* damped in a speed that is at least proportional to $(1 - \mu)^{q+k}$, leading to a form of hypergradient scarcity.

The rest of this section is dedicated to proving Lemma 4.1 and Theorem 4.2. We express $\boldsymbol{Y}(\boldsymbol{A})$ as a Neumann series, then we bound the derivative of terms in the resulted series, and by extension the hypergradient.

## 4.1 Neumann series expansion and the proof of Lemma 4.1

In the first step, we re-write the inverse of $\boldsymbol{B}$ using Neumann series. We first need to prove that $\|\boldsymbol{I} - \boldsymbol{B}\| < 1$ (see *e.g.*, Stewart (1998)), where $\boldsymbol{I} \in \mathbb{R}^{n \times n}$ is the identity matrix. Remark that the eigenvalues of $\boldsymbol{I} - \boldsymbol{B}$ are $1 - \mu_i$ where $\mu_1, \ldots, \mu_n$ are the eigenvalues of $\boldsymbol{B}$. Assuming the graph is connected, the ordered eigenvalues $\{\nu_i\}_{i=0}^n$ of $\tilde{\boldsymbol{L}}$ satisfy:

$$0 = \nu_1 < \nu_2 \leq \ldots \leq \nu_n \leq 2 \ . \tag{13}$$

The last inequality holds because $\|\boldsymbol{L}\| \leq 2d_{\max} \leq 2|E|$, where $d_{\max}$ is the maximum degree of the graph. Let $\boldsymbol{u}_1, \ldots, \boldsymbol{u}_n$ be the eigenvectors of $\tilde{\boldsymbol{L}}$, where $\boldsymbol{u}_1 \propto \boldsymbol{1}_n$ is associated to 0.

*Proof of Lemma 4.1.* We have $\|\tilde{\boldsymbol{S}}_{in}\| \leq 1/|V_{tr}|$ and $\|\tilde{\boldsymbol{L}}\| \leq 2$ so by a triangular inequality the upper bound is proved.

Using the eigendecomposition of $\tilde{\boldsymbol{L}}$ and recalling that $\nu_1 = 0$, for any $\boldsymbol{x} \in \mathbb{R}^n$:

$$\boldsymbol{x}^\top \boldsymbol{B} \boldsymbol{x} = \lambda \boldsymbol{x}^\top \tilde{\boldsymbol{L}} \boldsymbol{x} + \boldsymbol{x}^\top \tilde{\boldsymbol{S}}_{in} \boldsymbol{x}$$

$$= \lambda \sum_{i=2}^n (\boldsymbol{x}^\top \boldsymbol{u}_i)^2 \nu_i + \frac{\sum_{i \in V_{tr}} \boldsymbol{x}_i^2}{|V_{tr}|}$$

which, minimized over the unit sphere, gives the expression of $\mu_{\min}$. It is immediate that $\mu_{\min} \geq 0$. We prove that this value is strictly positive. Indeed, $\boldsymbol{x}^\top \boldsymbol{B} \boldsymbol{x} = 0$ implies that $\boldsymbol{x}^\top \boldsymbol{S}_{in} \boldsymbol{x} = 0$ and therefore $\boldsymbol{x}_i = 0$ for $i \in V_{tr}$, but also that $\boldsymbol{L}\boldsymbol{x} = 0$ and therefore that $\boldsymbol{x} \propto \boldsymbol{1}_n$, which implies that $\boldsymbol{x} = 0$. $\square$

Let $\tilde{\boldsymbol{B}} = \boldsymbol{B}/\mu_{\max}$, with eigenvalues $\mu_i/\mu_{\max}, i = 1, \ldots, n$. We have for all $i$:

$$0 \leq 1 - \mu_i/\mu_{\max} \leq 1 - \mu < 1 \ ,$$

where $\mu = \frac{\mu_{\min}}{\mu_{\max}}$. Therefore, we can use Neumann expansion to express $\tilde{\boldsymbol{B}}^{-1}$ as follows:

$$\tilde{\boldsymbol{B}}^{-1} = \sum_{r=0}^\infty (\boldsymbol{I} - \tilde{\boldsymbol{B}})^r \Rightarrow \boldsymbol{Y}(\boldsymbol{A}) = \sum_{r=0}^\infty (\boldsymbol{I} - \tilde{\boldsymbol{B}})^r \mu_{\max}^{-1} \tilde{\boldsymbol{S}}_{in} \boldsymbol{Y}_{obs} \ . \tag{14}$$

We denote by $\boldsymbol{T}_r$ the $r$-th term in $\boldsymbol{Y}(\boldsymbol{A})$:

$$\boldsymbol{T}_r = (\boldsymbol{I} - \tilde{\boldsymbol{B}})^r \mu_{\max}^{-1} \tilde{\boldsymbol{S}}_{in} \boldsymbol{Y}_{obs} \ . \tag{15}$$

Note that since $\|\boldsymbol{S}_{in}\boldsymbol{Y}_{obs}\| \leq \sqrt{|V_{tr}|}\|\boldsymbol{Y}_{obs}\|_\infty$, we have:

$$\|\boldsymbol{T}_r\| \leq \frac{\nu^r y_\infty}{\mu_{\max}\sqrt{|V_{tr}|}} \quad, \tag{16}$$

where $y_\infty = \|Y_{obs}\|_\infty$ and $\nu = 1 - \mu$. Similarly, $\|\boldsymbol{Y}(\boldsymbol{A})\| \leq \frac{y_\infty}{\mu_{\min}\sqrt{|V_{tr}|}}$. Moreover, since $\boldsymbol{I} - \tilde{\boldsymbol{B}}$ has the same zero-pattern than $\boldsymbol{A}$ (except on the diagonal), if $u$ is more than $r$ hops from $V_{tr}$, we get $(\boldsymbol{T}_r)_u = 0$.

### 4.2 Gradient of $(\boldsymbol{T}_r)_u$

In the second step, we derive the formula of the gradient of $(\boldsymbol{T}_r)_u$ w.r.t. $\boldsymbol{A}$, and derive a bound on its magnitude as a function of $r$, $q$ the distance to $V_{tr}$, and $k$ the distance to $V_{out}$. For $r > 0$, the gradient of the $u$-th coefficient in $\boldsymbol{T}_r$ w.r.t. $\boldsymbol{I} - \tilde{\boldsymbol{B}}$ is:

$$\nabla_{\boldsymbol{I}-\tilde{\boldsymbol{B}}}(\boldsymbol{T}_r)_u = \sum_{h=1}^{r}\left(\left((\boldsymbol{I} - \tilde{\boldsymbol{B}})^{r-h}\right)_{u,:}\right)^\top$$
$$\times \left((\boldsymbol{I} - \tilde{\boldsymbol{B}})^{h-1}\mu_{\max}^{-1}\tilde{\boldsymbol{S}}_{in}\boldsymbol{Y}_{obs}\right)^\top,$$

by the product rule of differentiation, and we have

$$\nabla_{\boldsymbol{I}-\tilde{\boldsymbol{B}}}(\boldsymbol{T}_r)_u = \sum_{h=1}^{r}\left(\left((\boldsymbol{I} - \tilde{\boldsymbol{B}})^{r-h}\right)_{u,:}\right)^\top (\boldsymbol{T}_{h-1})^\top \quad.$$

Using that $\boldsymbol{I} - \tilde{\boldsymbol{B}} = \boldsymbol{I} - \frac{1}{\mu_{\max}}(\tilde{\boldsymbol{S}}_{in} + \lambda\tilde{\boldsymbol{L}})$, we have

$$\nabla_{\tilde{\boldsymbol{L}}}(\boldsymbol{T}_r)_u = -\frac{\lambda}{\mu_{\max}}\nabla_{\boldsymbol{I}-\tilde{\boldsymbol{B}}}(\boldsymbol{T}_r)_u$$
$$= -\frac{\lambda}{\mu_{\max}}\sum_{h=1}^{r}\left(\left((\boldsymbol{I} - \tilde{\boldsymbol{B}})^{r-h}\right)_{u,:}\right)^\top (\boldsymbol{T}_{h-1})^\top \quad. \tag{17}$$

Finally, by deriving $\tilde{\boldsymbol{L}}$ w.r.t. $\boldsymbol{A}_{ij}$:

$$\frac{\partial(\boldsymbol{T}_r)_u}{\partial\boldsymbol{A}_{ij}} = -\frac{\lambda}{|E|\mu_{\max}}\sum_{h=1}^{r}\left((\boldsymbol{I} - \tilde{\boldsymbol{B}})^{r-h}\right)_{ui}(\boldsymbol{T}_{h-1})_i \tag{18}$$
$$+ \left((\boldsymbol{I} - \tilde{\boldsymbol{B}})^{r-h}\right)_{uj}(\boldsymbol{T}_{h-1})_j$$
$$- \left((\boldsymbol{I} - \tilde{\boldsymbol{B}})^{r-h}\right)_{uj}(\boldsymbol{T}_{h-1})_i$$
$$- \left((\boldsymbol{I} - \tilde{\boldsymbol{B}})^{r-h}\right)_{ui}(\boldsymbol{T}_{h-1})_j \quad,$$

which allows us to prove the following.

**Lemma 4.3.** *Let $i, j, u$ such that: $i, j$ are at least $k$-hop from $u$, and at least $q$-hop from $V_{tr}$. Then:*

$$\left|\frac{\partial(\boldsymbol{T}_r)_u}{\partial\boldsymbol{A}_{ij}}\right| \leq \begin{cases} 0 & \text{if } q + k > r \\ \frac{4\lambda y_\infty}{|E|\mu_{\max}^2\sqrt{|V_{tr}|}}(r - q - k)\nu^{r-1} & \text{otherwise.} \end{cases} \tag{19}$$

*Proof.* Recall that $(\boldsymbol{T}_r)_u = 0$ if $u$ is more than $r$-hop from $V_{tr}$. Similarly, $((\boldsymbol{I} - \tilde{\boldsymbol{B}})^r)_{ui} = 0$ if $u$ and $i$ are more than $r$-hop from each other. Hence, the term $\left((\boldsymbol{I} - \tilde{\boldsymbol{B}})^{r-h}\right)_{ui}(\boldsymbol{T}_{h-1})_i$ appearing in Eq. (18) equals 0 if $r - h < k$ or $h - 1 < q$, and bounded by $(\mu_{\max}\sqrt{|V_{tr}|})^{-1}\nu^{r-1}y_\infty$ otherwise. In a similar way for the other terms in Eq. (18), we get that the sum runs over the indices $h$ that satisfy $q + 1 \leq h \leq r - k$, which is either none if $q + k > r$, or $r - q - k$ terms otherwise, which concludes the proof. $\square$

### 4.3 Proof of Theorem 4.2

We finally examine the hypergradient, and prove an exponential damping rate of its magnitude with the cumulated distance to $V_{tr}$ and $V_{out}$.

**Definition 4.1** (Node cumulated distance to two subsets of nodes). The cumulated distance of a node $i$ to two subsets of nodes $V_{tr}, V_{out}$ equals $q + k$, where $q, k$ are the distances of $i$ to $V_{tr}, V_{out}$, respectively.

**Definition 4.2** (Edge cumulated distance to two subsets of nodes). The cumulated distance of an edge $(i, j)$ to two subsets of nodes $V_{tr}, V_{out}$ equals the minimum between the cumulated distance of $i$ and the cumulated distance of $j$ to the subsets $V_{tr}, V_{out}$.

Considering $F_{out} = \|\boldsymbol{S}_{out}(\boldsymbol{Y}(\boldsymbol{A}) - \boldsymbol{Y}_{obs})\|^2$, where $\boldsymbol{S}_{out}$ is the diagonal selection matrix whose diagonal entries equal 1 if the corresponding node is in $V_{out}$ and 0 otherwise, we have:

$$\frac{\partial F_{out}}{\partial \boldsymbol{A}_{ij}} = 2(\frac{\partial \boldsymbol{Y}(\boldsymbol{A})}{\partial \boldsymbol{A}_{ij}})^\top \boldsymbol{S}_{out}(\boldsymbol{Y}(\boldsymbol{A}) - \boldsymbol{Y}_{obs})$$

$$= 2\sum_{r=0}^{\infty}(\frac{\partial \boldsymbol{T}_r}{\partial \boldsymbol{A}_{ij}})^\top \boldsymbol{S}_{out}(\boldsymbol{Y}(\boldsymbol{A}) - \boldsymbol{Y}_{obs}) \ .$$

Using a triangular inequality, the bound on $\|\boldsymbol{Y}(\boldsymbol{A})\|$, and that $\|\boldsymbol{S}_{out}\boldsymbol{Y}_{obs}\| \le \sqrt{|V_{out}|}y_\infty$ we get:

$$\|\boldsymbol{S}_{out}(\boldsymbol{Y}(\boldsymbol{A}) - \boldsymbol{Y}_{obs})\| \le \frac{1 + \mu_{\min}\sqrt{|V_{tr}||V_{out}|}}{\mu_{\min}\sqrt{|V_{tr}|}}y_\infty \ .$$

By incorporating the resulting inequality in bounding the hypergradient, and by noticing that $\boldsymbol{S}_{out} = \boldsymbol{S}_{out}^2$ we have:

$$\left|\frac{\partial F_{out}}{\partial \boldsymbol{A}_{ij}}\right| \lesssim \frac{1 + \mu_{\min}\sqrt{|V_{tr}||V_{out}|}}{\mu_{\min}\sqrt{|V_{tr}|}}y_\infty \sum_{r=0}^{\infty}\|\boldsymbol{S}_{out}\frac{\partial \boldsymbol{T}_r}{\partial \boldsymbol{A}_{ij}}\|$$

$$\lesssim \frac{1 + \mu_{\min}\sqrt{|V_{tr}||V_{out}|}}{\mu_{\min}\sqrt{|V_{tr}|}}y_\infty \sum_{r=0}^{\infty}\left(\sum_{u \in V_{out}}\left|\frac{\partial (\boldsymbol{T}_r)_u}{\partial \boldsymbol{A}_{ij}}\right|^2\right)^{\frac{1}{2}} \ .$$

Using Lemma 4.3 and the hypotheses on $i$ and $j$, for $u$ in $V_{out}$, the term $\left|\frac{\partial (\boldsymbol{T}_r)_u}{\partial \boldsymbol{A}_{ij}}\right|$ is 0 if $r < q + k + 1$, and bounded by $\frac{4\lambda y_\infty}{|E|\mu_{\max}^2\sqrt{|V_{tr}|}}(r - q - k)\nu^{r-1}$ otherwise. Hence:

$$\left|\frac{\partial F_{out}}{\partial \boldsymbol{A}_{ij}}\right| \lesssim \lambda\frac{\sqrt{|V_{out}|} + \mu_{\min}\sqrt{|V_{tr}|}|V_{out}|}{\mu_{\min}|V_{tr}||E|\mu_{\max}^2}y_\infty^2$$

$$\times \sum_{r=q+k+1}^{\infty}(r - q - k)\nu^{r-1} \ .$$

Then we see that for $\nu < 1$ we have

$$\sum_{r=q+k+1}^{\infty}(r - q - k)\nu^{r-1} = \nu^{q+k}\sum_{r=1}^{\infty}r\nu^{r-1} \ ,$$

and $\sum_{r=1}^{\infty}r\nu^{r-1} = \frac{1}{(1-\nu)^2} = \frac{1}{\mu^2}$, which concludes the proof.

## 5 Alleviating hypergradient scarcity

In this section, we review strategies to mitigate hypergradient scarcity. However, it is important that we make a distinction between resolving this issue and resolving the overfitting problem. Indeed, if gradient

scarcity is also caused by the limited quantity of available labeled data, it is important to avoid confusion with traditional overfitting. In particular, while traditional overfitting is generally reduced by adding more training data, *gradient scarcity is still observed when optimizing edges far from labeled nodes regardless of the dataset size and the number of labels.* We study several strategies to mitigate hypergradient scarcity in the bilevel setting, but we emphasize that they might not lead to a better generalization error altogether.

**Generalized edge refinement by optimizing $A_{obs}^r$.** As hypergradient scarcity is observed on edges connecting nodes distant from the labeled ones, a natural fix is to reduce this distance. One way to do that is by refining edges in a power of $A_{obs}$, as the matrix $A_{obs}^r$ includes $r$-edge long connections between nodes. In our experiments, we adopt $A_{obs}^6$ as this notably expands the graph but does not achieve the extreme case where the result is a complete graph.

**Graph regularization.** Graph regularization is used to impose a prior structure on the learned graph, by adding a regularization term to $F_{out}$ to penalize graphs with undesirable properties. For instance, Kalofolias (2016) propose the regularization term $-\gamma \mathbf{1}_n^\top \log A \mathbf{1}_n$ for some $\gamma > 0$ to penalize low-degree nodes. We use this choice in the experiments, but note that imposing task-related priors and regularization terms could lead to better performance. This will be the topic of future work.

**G2G for edge refinement.** The third fix we suggest is latent graph learning using G2G models. In the outer problem, we propose to replace optimizing edge weights by optimizing the parameters of a G2G model to predict similarity between nodes. Let $\theta$ be the weights of this model, and $A_\theta$ be its output graph, the G2G model we adopt is $(A_\theta)_{i,j} = \alpha\big((X_i - X_j)^2\big)$, where the square function is applied entrywise, $\alpha : \mathbb{R}^p \to \mathbb{R}$ is a Multi-Layer Perceptron (MLP) model consisting of $k_{G2G}$ layers, each is of the form:

$$X^{[l]} = \phi^{[l]}\big(X^{[l-1]} W_1^{[l]} + \mathbf{1}_n (b^{[l]})^\top\big) \ ,$$

where $W_1^{[l]} \in \mathbb{R}^{d_{l-1} \times d_l}, b^{[l]} \in \mathbb{R}^{d_l}$ are learnable parameters, and $d_l$ is the output dimensionality of the $l$-th layer. The parameters are gathered as $\theta = \{W_1^{[l]}, b^{[l]}\}_{l=1}^{k_{G2G}}$.

# 6 Experiments

We[1] use the real-world citation networks Cora (Lu & Getoor, 2003), CiteSeer (Bhattacharya & Getoor, 2007), and PubMed (Namata et al., 2012), and two other synthetic datasets to validate our findings. The first synthetic dataset, called synthetic dataset 1, is designed for the Laplacian regularization scenario. The second one is a binary classification dataset that can be used for the three considered graph-based models. Due to the paradigm behind construction, we call it the cheaters dataset.

**Synthetic dataset 1:** the purpose of this dataset is to validate our findings on regression problems, particularly with the Laplacian regularization 2 in the inner problem. We sample *i.i.d.* latent variables $X \in \mathbb{R}^{n \times p}$ as the node features uniformly at random from $[0, 1]$ with $n = 1536, p = 2$. The ground-truth graph $A^\star$ is constructed s.t. $(A^\star)_{i,j} = 1$ if $\|X_i - X_j\|_2 < \sigma$, and 0 otherwise. $\sigma$ is set to 0.06 s.t. the number of edges approximates $n \log n$. Two distinct procedures were employed to sample the nodes that comprise $V_{tr}$, leading to two distinct realizations of the dataset as illustrated in Fig. 1(top). The first procedure randomly samples 100 nodes from the set $V$, hence $V_{tr}$ is well-spread, whereas the second procedure selects the 100 nodes with the smallest Euclidean distance to the point $(0.5, 0.5)$, thus $V_{tr}$ is concentrated in a small neighborhood in this case. In both cases, we randomly sample 25 nodes from $V$ to construct $V_{out}$. The remaining nodes are equally divided between the validation and the test sets. Then, each node $i$ in $V_{tr}$ is labeled as follows:

$$(Y_{obs})_i = \zeta\big(e^{-\frac{\|X_i - a_1\|^2}{2(0.2)^2}} + e^{\frac{-\|X_i - a_2\|^2}{2(0.2)^2}} + e^{\frac{-\|X_i - a_3\|^2}{2(0.2)^2}}\big) \ ,$$

where $a_1, a_2, a_3$ are randomly sampled from $[0, 1]^2$, and $\zeta$ is a scaling factor such that labels lie in $[0, 1]$. By this construction, the prior that the labeling function on the graph is smooth is met, and the Laplacian regularization can be applied as in Eq. (1).

---

[1]Our *Python* implementation is available at `https://github.com/hashemghanem/Gradients_scarcity_graph_learning`.

We use the Laplacian regularization model to generate labels for other nodes. That is, we plug the labels of $V_{tr}$ and $\boldsymbol{A}^\star$ in Eq. (1) with $\lambda = 1$ and the solution holds the sought-for labels. This way, the ground-truth graph actually plays a role in labeling nodes in $V_{out}$ and in the validation set.

The noisy observed graph is built upon random weights

$$(\boldsymbol{A}_{obs})_{i,j} = \xi_{i,j}(\boldsymbol{A}^\star)_{i,j} \quad \text{where} \quad \xi_{i,j} \sim \mathcal{U}([0,1]) \ .$$

Experiments on this dataset are done with the Laplacian regularization in the inner problem as in Eq. (1).

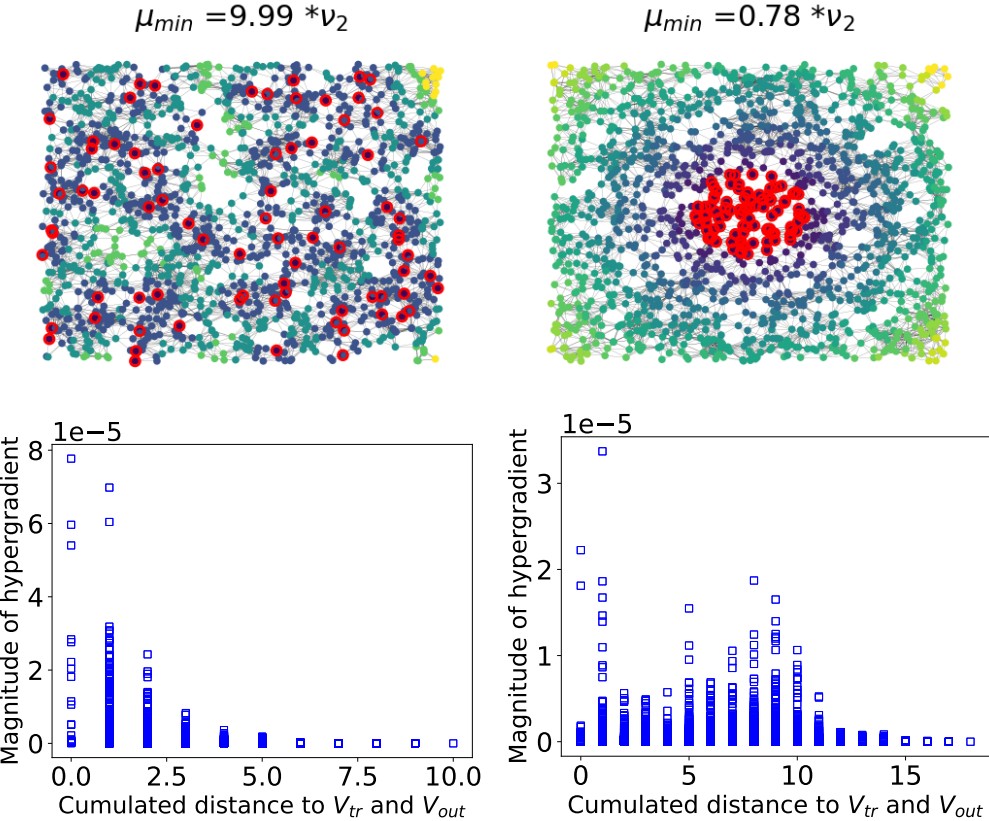

Figure 1: Hypergradient scarcity observed when solving the edge refinement task with the bilevel optimization framework. We run the experiments on the synthetic dataset 1, while adopting the Laplacian regularization in the inner problem. **Top**: illustration of the graph. The training nodes $V_{tr}$ are circled in red, the colors correspond to the distance to $V_{tr}$. The eigenvalue $\mu_{\min}$ is given as a ratio of the smallest positive eigenvalue of $\tilde{\boldsymbol{L}}$. $V_{out}$ is randomly sampled from $V$ but not shown here. **Bottom**: Hypergradient magnitude $\left|\frac{\partial F_{out}}{\partial \boldsymbol{A}_{ij}}\right|$ with respect to the *sum* of distances to $V_{tr}$ and $V_{out}$. **Left**: the training set $V_{tr}$ is well-spread thereby aligned with the high-frequency eigenvectors of the graph, resulting in a *high* $\mu_{\min}$. The decrease of the hypergradients is sharp with the distance. **Right**: $V_{tr}$ is aligned with the low-frequency eigenvectors of the graph, resulting in a *low* $\mu_{\min}$. The decrease of hypergradients magnitude is not as sharp as the previous case.

**The cheaters dataset:** the purpose of this dataset is to validate our findings on classification problems, particularly with the shallow GNN 4 in the inner problem, and to validate the efficiency of the proposed strategies to alleviate hypergradient scarcity. Nodes in this graph represent students in an exam classroom. Setting $n = 256, p = 10$, the *i.i.d.* features $\boldsymbol{X} \in \mathbb{R}^{256 \times 10}$ are sampled uniformly at random from $[0,1]$. For a node $i$, $\boldsymbol{X}_{i,0}$ represents the position of the according student in the classroom. For visualization purposes, we enumerate nodes following the ascending order of $\boldsymbol{X}_{:,0}$. The remaining 9 features of a student represent the grades s/he is capable of scoring in the corresponding exam questions. However, students tend to cheat

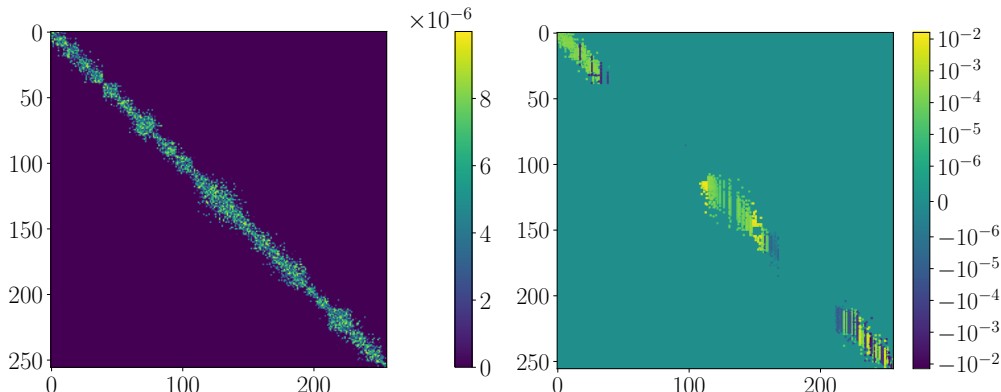

Figure 2: Hypergradient scarcity observed when solving the edge refinement task with the bilevel optimization framework. We run the experiment on the cheaters dataset, and use a 2-layer GNN as a classifier following the model 4. Left: graph initialization. Right: hypergradient at an arbitrary outer iteration, namely 9. It is clear that the hypergradient on edges between unlabeled nodes far from the ones in $V_{out} \cup V_{tr}$ equals zero. Recall that $V_{tr} = \{0, 1, \dots, 32\} \cup \{224, \dots, 255\}$ and $V_{out} = \{96, \dots, 160\}$.

with their neighbors in the graph. The ground-truth graph $\boldsymbol{A}^\star$ is constructed as follows:

$$(\boldsymbol{A}^\star)_{i,j} = \exp\left(-\|\boldsymbol{X}_{i,0} - \boldsymbol{X}_{j,0}\|_2^2 / 2\sigma^2\right) .$$

The observed graph $\boldsymbol{A}_{obs}$ is drawn from a random model as

$$(\boldsymbol{A}_{obs})_{i,j} \sim \mathrm{Ber}\left((\boldsymbol{A}^\star)_{i,j}\right).$$

We set $\sigma = 0.027$ s.t. the number of edges in $\boldsymbol{A}_{obs}$ approximates $n \log n$. Students cheat such that their grades $\boldsymbol{Y}_{grade}$ after the exam are

$$\boldsymbol{Y}_{grade} = \boldsymbol{A}^\star \boldsymbol{X}_{:,1:9} \boldsymbol{1}_9 .$$

A student passes the exam if his grade is greater than a threshold $\tau$, *i.e.,* $(\boldsymbol{Y}_{obs})_i = 1$ if $(\boldsymbol{Y}_{grade})_i > \tau$ and 0 otherwise. We put $\tau = 60$ so that approximately half of the students pass the exam. $V_{tr}$ includes nodes in $\{0, 1, \dots, n/8\} \cup \{7n/8, \dots, n-1\}$, *i.e.,* near the two ends of the 1-dimensional class. $V_{out} = \{3n/8, \dots, 5n/8\}$, *i.e.,* centered around the middle of the class. Remaining nodes are equally divided into a validation and a test set. Experiments on this dataset are done with a 2-layer GNN classifier following the model 4.

**Real-world dataset:** we validate our findings on the Cora (Lu & Getoor, 2003), CiteSeer (Bhattacharya & Getoor, 2007), and PubMed (Namata et al., 2012) datasets. These are citation datasets, where nodes represent research publications described by a bag of words, and edges stand for citations. The task is to classify articles *w.r.t.* their topic. From the default train/validation/test split in Yang et al. (2016); Kipf & Welling (2017), we use the training set as the inner training set $V_{tr}$, while we use half of the validation set as the outer training set $V_{out}$. The other half is kept as a validation set as in Franceschi et al. (2019).

**Models:** G2G and GNN models are implemented using *PyTorch* (Paszke & al., 2019) and *PyTorch* Geometric (Fey & Lenssen, 2019), respectively. The function $\alpha$ in the G2G model is an MLP with 2 hidden layers of 16 neurons for the cheaters dataset, and 1 hidden layer of 32 neurons for the citation datasets. The GNN model following Eq. (4) has 1 hidden layer of 8 neurons for the cheaters dataset and 16 neurons for the citation datasets. To compute the node embeddings for the APPNP model, we use an MLP with one hidden layer. The hidden and the output layers contain 16 neurons. For the APPNP model, the number of power iterations is set to 20 and the teleport probability is set to 0.1. All hidden layers are followed by the *ELU* activation function (Clevert et al., 2015). The G2G output layer is followed by the sigmoid function. The output layer of GNN models is followed by the *softmax* function.

**Setup:** we use Adam (Kingma & Ba, 2014) as the inner and the outer optimizer with the default parameters of *PyTorch*, except for the inner learning rate $\eta_{in}$ and the outer one $\eta_{out}$, which are tuned from the set

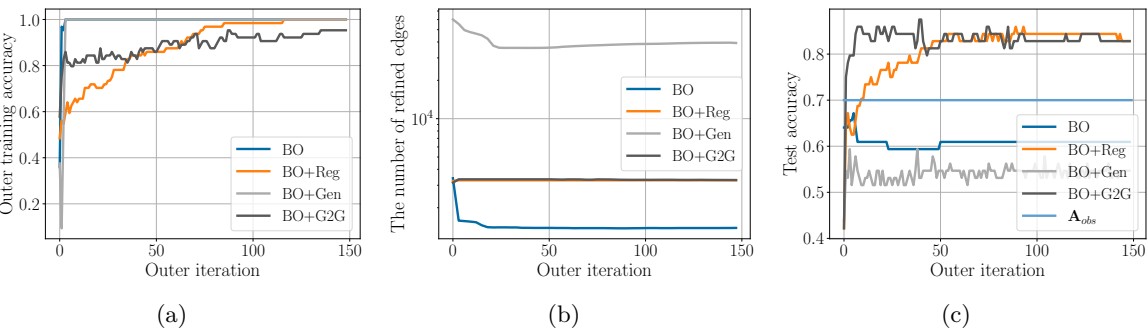

Figure 3: Efficiency of the proposed solutions to hypergradient scarcity *w.r.t.* the number of refined edges and the generalization capacity. An edge is considered well refined if its learned magnitude is larger than one percent of the maximum learned edge weight. The solutions are graph regularization with $-\mathbf{1}^\top \log \boldsymbol{A}\mathbf{1}$ (BO+Reg), generalized edge refinement by refining edges in $\boldsymbol{A}_{obs}^6$ (BO+Gen), and latent graph learning using a G2G model (BO+G2G). (a): training accuracy on $V_{out}$. (b): number of refined edges. (c) test accuracy.

$\{10^{-4}, 10^{-3}, \dots, 10\}$. The best values were $\eta_{in} = \eta_{out} = 10^{-2}$ for the citation datasets. For the cheaters dataset, $\eta_{out} = 10^{-3}$ adopting a G2G model, while $\eta_{out} = 10^{-2}$ in other cases, and $\eta_{in} = 10^{-2}$. For the synthetic dataset 1, $\eta_{out} = 10^{-1}$ and $\eta_{in} = 10$. We set $\tau_{in}$ with a grid search. For the citation datasets, $\tau_{in} = 500$ adopting the Laplacian regularization, and $\tau_{in} = 100$ otherwise. For the cheaters dataset, $\tau_{in} = 200$. For the synthetic dataset 1, $\tau_{in} = 500$. The learnable parameters of the inner model are initialized at random after each outer iteration. We adopt the default initialization of *PyTorch* and *PyTorch* Geometric for the parameters of the GNN models. In the Laplacian regularization scenario, the labels are initialized uniformly at random from $[0, 1]$. Except for the cheaters dataset, we initialize edge weights uniformly at random from $[0, 1]$ when solving an edge refinement task, and adopt the default initialization of *PyTorch* for the G2G model when it is used. For the cheaters dataset, we initialize edge weights uniformly at random from $10^{-5} * [0, 1]$ in edge refinement tasks, and multiply the default initialization of the last layer of the G2G model by $10^{-5}$ s.t. its output edges at the first iteration are of small magnitude. We adopt this strategy for this dataset to measure the level of hypergradient scarcity, which we do by counting the number of learned edges whose magnitude is greater than a chosen threshold. We set the number of outer iterations $\tau_{out}$ to 150 for synthetic datasets and to 300 for citation datasets. We select the graph (or the G2G parameters) with the highest validation accuracy. We set $\lambda = 1$ in training when considering the Laplacian regularization, as we expect the bilevel framework to learn this parameter by scaling the learned adjacency matrix. When applying the Laplacian regularization fed with $\boldsymbol{A}_{obs}$ on the citation datasets, we set $\lambda = 1$ after a grid search. $\gamma$ in the graph regularization term is set to 1 following a grid search on the set $\{10^{-3}, 10^{-2}, \dots, 10\}$.

### 6.1 Hypergradient scarcity with the GNN model in Eq. (4)

In this experiment, we consider a 2-layer GNN classifier following the model 4 in the bilevel framework. We solve the edge refinement task 6 on the cheaters dataset, where $\ell$ in Eqs. (1) and (6) is the CCE function. Fig. 2(left) depicts the initialization of the adjacency matrix. It also shows what edges are to be optimized, that is, edges whose initialization is nonzero. In Fig. 2(right), we show the hypergradient at the outer iteration 9, which is arbitrarily chosen. It is clear that edges between unlabeled nodes far from the ones in the union $V_{out} \cup V_{tr}$ get no supervision during the training process. Recall that $V_{tr} = \{0, 1, \dots, 32\} \cup \{224, \dots, 255\}$ and $V_{out} = \{96, \dots, 160\}$. This aligns with our findings, which state that edges between nodes at least 2-hop from nodes in $V_{out} \cup V_{tr}$ receive zero hypergradients. This, as seen in Fig. 3, leads to a learned graph that overfits training nodes and even generalizes worse than $\boldsymbol{A}_{obs}$.

Table 1: Accuracies obtained on citation networks with the Bilevel Optimization framework (BO), the same framework optimizing a G2G model (BO+G2G), and the same framework equipped with graph regularization (BO+Reg). We also benchmark against GAM (the result is reported from the according paper) and against $\boldsymbol{A}_{obs}$. Each experiment is repeated 5 times and the average accuracies are reported. For each dataset, the first line (in black) corresponds to the test accuracy, whereas the second line (in gray) corresponds to the training accuracy on $V_{out}$. The highest and second-highest test accuracies for each dataset are bolded. The training accuracy on $V_{tr}$ is higher than 96% in all experiments.

| | $\boldsymbol{A}_{obs}$ | | | BO | | | BO+G2G | | | BO+Reg | | | GAM |
|---|---|---|---|---|---|---|---|---|---|---|---|---|---|
| | GNN | Lap | APPNP | GNN | Lap | APPNP | GNN | Lap | APPNP | GNN | Lap | APPNP | |
| Cora | 73.62 | 72.40 | 80.88 | 79.07 | 73.80 | 79.90 | 80.32 | 75.76 | 79.96 | 80.76 | 76.45 | **82.10** | **84.8** |
| | 77.50 | 74.03 | 77.78 | 90.17 | 81.70 | 83.73 | 94.58 | 83.64 | 77.66 | 97.30 | 84.68 | 87.80 | - |
| CiteSeer | 62.48 | 54.00 | 70.08 | 68.10 | 53.00 | 71.04 | 70.80 | 54.72 | **71.82** | 70.60 | 55.10 | 71.50 | **72.20** |
| | 59.66 | 56.28 | 68.09 | 75.94 | 57.74 | 76.92 | 72.76 | 57.90 | 81.44 | 74.09 | 61.96 | 70.42 | - |
| PubMed | 77.42 | 70.50 | 79.14 | 77.68 | 72.80 | 79.32 | 79.00 | 72.18 | 79.22 | 78.40 | 75.75 | **80.20** | **81.00** |
| | 79.58 | 74.71 | 80.36 | 93.47 | 80.00 | 85.38 | 90.27 | 71.37 | 94.64 | 87.59 | 83.39 | 85.38 | - |

## 6.2 Hypergradient scarcity with the Laplacian regularization model

We here examine hypergradient scarcity when adopting the Laplacian regularization in the inner problem. We run the bilevel optimizer to solve the edge refinement task on the synthetic dataset 1. The dataset corresponds to a regression problem, so $\ell$ in Eqs. (1) and (6) is the MSE loss function.

In Fig. 1(bottom), we plot the absolute value of hypergradients at the outer iteration 6 as a function of the edge cumulated distance to $V_{tr}$ and $V_{out}$. One observes the hypergradient scarcity phenomenon, since hypergradients decay exponentially as the edge distance increases. This validates our analysis articulated in Theorem 4.2. In addition, we observed in practice that $\mu$ is nevertheless quite small, and that our bound in Theorem 4.2 is quite loose. Another observation is that the decrease rate is higher when $V_{tr}$ is well-spread in the graph. Deriving a tighter bound on the magnitude of hypergradients and investigating the link between the distribution of labeled nodes and this bound will be the subject of a future work.

## 6.3 Testing solutions to mitigate hypergradient scarcity

We run our experiments on the cheaters dataset using the 2-layer GNN 4 as a classifier. In each experiment, we run the bilevel optimization framework with one of the suggested fixes. We consider two criteria to measure the efficiency of each solution, the first one is counting the number of refined edges. At any outer iteration, we say that an edge is refined if its learned weight is greater than one percent of the maximum learned edge weight at the same iteration. Recall that we initialize the graph/G2G with small parameters ($\approx 10^{-5}$). The second criterion is the test accuracy. The first criterion assesses the ability to alleviate hypergradient scarcity, while the second assesses the generalization to nodes unseen during training, and thus if the learned graph is meaningful.

Fig. 3 shows that all three fixes produce better results *w.r.t.* the first criterion, as the number of refined edges is larger at almost every iteration, with optimizing edges in $\boldsymbol{A}_{obs}^{6}$ being the most efficient, and the G2G model and graph regularization having a similar performance. Moreover, one notices that this number decreases with the iteration index when refining edges in $\boldsymbol{A}_{obs}$ or in $\boldsymbol{A}_{obs}^{6}$, which is expected as only a small portion of edges receive supervision; however this portion is larger when refining $\boldsymbol{A}_{obs}^{6}$.

Regarding the second criterion, the G2G model and the graph regularization generalize well, as both combat hypergradient scarcity without increasing (or even by decreasing) the number of parameters to learn. On the other hand, optimizing edges in $\boldsymbol{A}_{obs}^{6}$ deteriorates performance in the test phase. A likely explanation is that by expanding the graph, we increase the number of parameters to learn, which means a more complex model that is more likely to overfit training nodes. This experiment illustrates that **hypergradient scarcity is not the traditional overfitting** related to data/label scarcity, and resolving it does not necessarily promote better generalization.

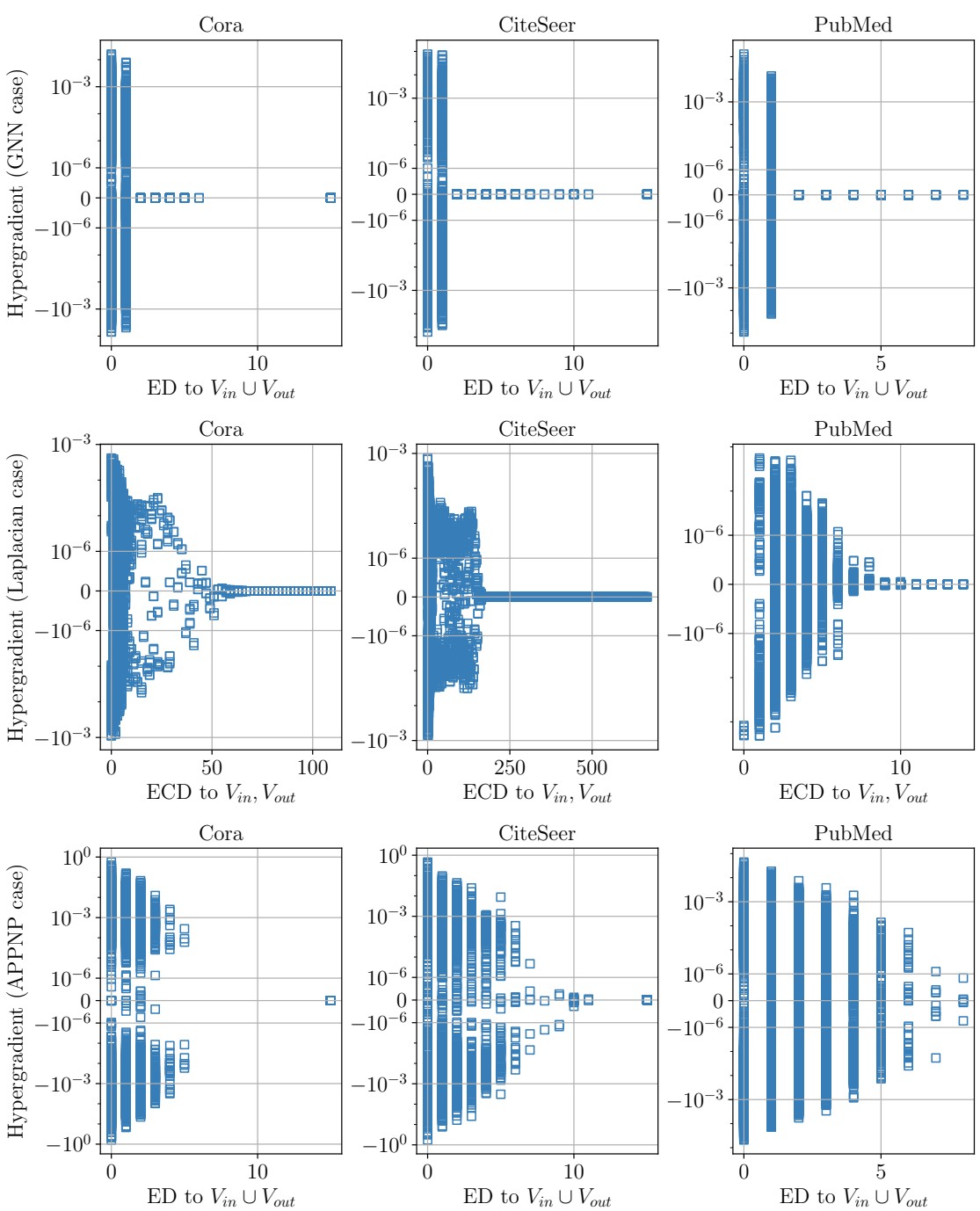

Figure 4: Hypergradient scarcity observed with citation networks when solving the edge refinement task under the bilevel optimization setting. **Top**: we depict the hypergradient as a function of the Edge Distance (ED) to $V_{in} \cup V_{out}$ when adopting the 2-layer GNN in Eq. (4) as the graph-based model. **Middle**: we depict the hypergradient as a function of the Edge Cumulated Distance (ECD) to the sets $V_{in}, V_{out}$ when adopting the Laplacian regularization model. Since our analysis in this scenario necessitates the graph to be connected, we make the graph connected by iterating over its components, and connecting an arbitrary node in the current component to an arbitrary node in the next one. The number of added edges is 77 for Cora, 437 for CiteSeer, and 0 for PubMed. This is done only in the experiment dedicated for this figure. In the other experiments, the given graph is considered. **Bottom**: we depict the hypergradient as a function of the Edge Distance (ED) to $V_{in} \cup V_{out}$ when adopting the APPNP model. In all cases, we plot the hypergradient at the outer iteration 9.

### 6.4 Results on real-world citation datasets

We use the bilevel optimization 6 for edge refinement on Cora, CiteSeer, and PubMed. The downstream task is a multi-label classification problem hence $\ell$ is the CCE function. Fig. 4 depicts the hypergradient on edges at outer iteration 9 as a function of their distance to labeled nodes. For the Laplacian regularization case, that is the edge cumulated distance to $V_{tr}$ and $V_{out}$. In the case of the GNN model in Eq. (4), the edge distance is obtained by computing the distance of its endpoints to $V_{tr} \cup V_{out}$ then taking the minimum. For APPNP, we do not have any theorem that defines what distance metric to use to observe hypergradient scarcity. We thus use the same metric as in the previous GNN case. In accordance with our analysis, the figure displays a null hypergradient for distances greater than 2 when using the GNN model in Eq. (4), while the Laplacian regularization scenario exhibits a hypergradient that diminishes exponentially with edge distance. Interestingly, APPNP exhibits a similar behavior to the Laplacian regularization model, where the hypergradient is damped exponentially with the edge distance to $V_{tr}$ and $V_{out}$. This means that although to a lesser degree than shallow GNNs, **GNNs of infinite receptive field such as APPNP suffer from hypergradient scarcity**, which is a new finding to the best of our knowledge.

Regarding the generalization capacity evaluated using the test accuracy, Table 1 shows that the bilevel framework (BO) outperforms $\boldsymbol{A}_{obs}$ when using the GNN model in Eq. (4), while it performs more or less the same when using the Laplacian regularization and APPNP models. This might be surprising as one would expect the bilevel framework to produce a more corrupted graph than $\boldsymbol{A}_{obs}$ due to hypergradient scarcity, and thus would lead to a worse generalization, which is observed with the cheaters dataset in Fig. 3. One possible justification for this is that node features are informative enough in these datasets to compensate for the corrupted graph. Another justification, especially in the case of the Laplacian regularization and APPNP, is the small diameter of the graph in these datasets, which limits the number of affected edges. For example, the percentage of edges with edge distance at most 2 is $77\%, 63\%, 35\%$ for Cora, CiteSeer, and PubMed, respectively. For edge distance at most 3, the percentages are $95\%, 80\%, 86\%$. This means that a great portion of edges are not affected by hypergradient scarcity. Hence, it is normal that the learned graph perform similarly or better than the observed one. Conducting experiments on other datasets where the graph is of higher importance and of larger diameter is left for future work.

Next, we test the efficiency of the proposed solutions to hypergradient scarcity *w.r.t.* test accuracy, namely graph regularization and latent graph learning with G2G models. We do not consider learning a power of $\boldsymbol{A}_{obs}$ as the memory requirement goes beyond the limits we have access to. Table 1 shows that both fixes yield significant improvements over $\boldsymbol{A}_{obs}$. More importantly, both fixes consistently improve over the bilevel framework, notably the graph regularization fix. Since the impact of hypergradient scarcity is not severe in these datasets, we cannot claim that the improvement is due to alleviating hypergradient scarcity. However, we can claim that the fixes do not deteriorate the generalization capacity, and that they are more robust to overfitting than the bilevel framework. Designing a metric to measure the severity of hypergradient scarcity and the impact of the fixes on it is left for future work. We point out that the bilevel optimization framework with either fix does not achieve state-of-the-art results produced by GAM.

Finally, we notice that GNN models lead to significantly superior results in all scenarios. This is expected, as the Laplacian regularization promotes similarity between connected nodes but, unlike GNNs, is not a supervised-based method.

## 7 Conclusion

We studied hypergradient scarcity when deploying bilevel optimization in edge refinement tasks under the semi-supervised learning setting. This phenomenon consists in edges far from labeled nodes receiving zero hypergradients when optimizing the graph and the graph-based model to improve the labeling performance. We proved that this problem occurs for shallow GNN models. Replacing GNNs by the Laplacian regularization model does not resolve the issue; however, the phenomenon is less severe. We bounded the magnitude of hypergradients and proved that they are exponentially damped with distance to labeled nodes. To alleviate hypergradient scarcity, we examined graph regularization, latent graph learning, and refining edges in a power of the observed adjacency matrix. Our experiments validated our findings, privileged the first two

solutions over the latter, and showed that GNNs of infinite receptive field (*e.g.,* APPNP) can suffer from hypergradient scarcity. Moreover, we showed that alleviating hypergradient scarcity does not necessarily alleviate overfitting.

**Acknowledgments**

The authors acknowledge the support of ANR Grava ANR-18-CE40-0005 and ANR GRandMa ANR-21-CE23-0006.

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
