# OpenReview forum: "Gradient Scarcity in Graph Learning with Bilevel Optimization"
_TMLR — Accepted by TMLR_

### Review · Reviewer_pERT · 2024-04-12

**Summary Of Contributions:**

This article focuses on gradient scarsity (specifically, the phenomenon where edges between two nodes cannot be learned if these two nodes are too far from the nodes in the training set because they receive zero gradient) and proposes several contributions. Firstly, it explains that this problem is linked to the fact that Graph Neural Networks (GNNs) have a finite receptive field. Secondly, it theoretically demonstrates the problem of gradient sparsity for two graph structure learning models: joint optimization and bilevel optimization. In the latter case, the contribution is twofold: first, when the inner problem involves predicting labels using a GNN, and second, when the inner problem involves predicting labels through propagation with Laplacian regularization. In this latter case, the authors show that the gradient is not zero but decreases exponentially fast. Furthermore, it empirically demonstrates that this problem also occurs for models with larger receptive fields.

**Audience:**

Yes

**Claims And Evidence:**

Yes

**Requested Changes:**

- Suggestions for Section 1.2:

When defining the bilevel problem, I believe it would be helpful to add some comments indicating that the problem resolution is impossible, and that in practice, a fixed number of gradient descent iterations is used for the inner problem. This would better justify the use of gradient descent (eq (7)) for Theorem 3.2 later on. I find eq (7) a bit out of context if one is not too familiar with bilevel problem resolution. This discussion comes too late in the "Bilevel optimization" paragraph of the experiment section.

- Confusion between weights and GNN weights:

I suggest referring to the weights of the GNN as "parameters" in the paper to distinguish them from the weights of the graph. There are a few places where this could lead to confusion or at least require some mental gymnastics that could be avoided (especially "In this first result, we will assume that the weights $W$ do not depend on A, which is the case in joint optimization").

- Regarding Theorem 3.1:

This theorem is very interesting. It already formally proves that the classical case exhibits gradient scarsity. However, there are three points that deserve discussion:

    - The joint problem $\min_{A,W}$ does not have any particular regularization on $A$: this does not seem very standard to me. I believe the theorem could be easily generalized to any differentiable regularization on the adjacency matrix ($A$ would still not depend on $W$).
    - I think the fact that $A$ does not depend on $W$ deserves some explanation. Simply stating that, ideally, we minimize a certain $G(W)$ where $G(W) = \min_A F(A, W)$ and thus $W$ does not depend on $A$.
    - In relation to the previous point: it seems to me that this is still an idealized case; in practice, we sometimes do alternating minimization, so, if I'm not mistaken, $W$ depends on the current ‘‘$A$''. Can the authors discuss this point?
    - I find the distinction between $k$-hop and $k$-layer a bit confusing: does the theorem really require that the nodes be at $k$-hop where $k$ is exactly the number of layers?

- Regarding Theorem 3.2:

In the proof, I think it would be beneficial to explain the chain rule a bit more to pedagogically explain that "By the chain rule, proving that [...] is sufficient to complete the proof."

- Regarding the experiments:

Overall, I find the experiments convincing. However, I think they lack discussion. I particularly think of the "Synthetic dataset 1" experiment: it is presented without describing its purpose or conclusion. We understand that the phenomenon of gradient scarsity is accentuated when the nodes in $V_{tr}$ are less concentrated. But in the second case, "The decrease of hypergradients magnitude is not as sharp as the previous case" is not discussed, for example. Why is this the case?

There are also some ambiguities in the definitions: "the distance to $V_{tr}$" is ambiguous (sum of the distances? average distance? minimal distance? we understand that it is the cumulative distance with the legend, but the text is a bit ambiguous). I think a formal definition would help clarify this. The same goes for the "Edge distance to $V_{tr} \cup V_{out}$".

- Other small remarks:
  - In the abstract, I find the sentence "It consists in edges between unlabeled nodes that are far from the labeled ones receiving zero gradient" for describing the gradient scarsity not very clear. I think it should be reformulated.
  - I think $\theta$ should be $W$ at the end of page 3.
  - I think $(1-u)^{q+k}$ should be $(1-\mu)^{q+k}$ on page 8.
  - In Figure 4: why display the gradient instead of just its norm as in Figure 1? (this unnecessarily overloads the figure)."

**Strengths And Weaknesses:**

Overall, I find this work ambitious and with many very interesting contributions. Firstly, the addressed problem is quite challenging, and the authors provide a clear response to it. The theorems are understandable (though not necessarily straightforward; they require significant formalization effort, blending bilevel optimization and graph theory). Additionally, I find the experiments convincing; they pinpoint a problem, which is the gradient scarsity even for models with larger receptive fields. Moreover, the authors have made an effort to make the article well-written. The related work section is also very comprehensive, which is pleasing.

For all these reasons, I am convinced that this work is of interest to the TMLR community and the broader ML community.

However, I have some remarks/suggestions on certain points that seem unclear to me and which, I think, should be further detailed (see below).

---

> ### Author Response · Authors · 2024-05-15
>
> We would like to thank the reviewer for the very detailed feedback and the constructive comments.
>
> _Concerning section 1.2._ We accordingly modified the organization by moving the discussion on the resolution of the bilevel problem from the experiment section to the introduction after defining the bilevel problem.
>
> _Concerning the weights and GNN weights._ We will refer to the GNN weights under the name "parameters".
>
> _Regarding Comment 1 on Theorem 3.1._ We thank the reviewer for this comment.
> Indeed, it is standard to adopt graph regularization in the joint setting, as in Fatemi et al. (2021) and Liu et al. (2022).
> In fact, Fatemi et al. (2021) showed that gradient scarcity emerges when the loss function is a labelling loss, i.e. in the absence of a regularization term.
> Then they showed that regularization can also mitigate gradient scarcity, not only overfitting.
> Given that we are interested in studying gradient scarcity, we considered no regurlarization terms, which is the setting in which this problems emerges.
> Another reason is that in the literature of bilevel frameworks mentioned above, it's not standard to consider graph regularization. So not considering it in the joint setting makes our study coherent and homogenous to the reader.
> However, we agree that it is important to show that the theory can be easily generalized to any differentiable regularization on the adjacency matrix. We will mention that directly after the theorem as an extension so the reader is aware of this fact.
>
> Fatemi, Bahare, Layla El Asri, and Seyed Mehran Kazemi. "Slaps: Self-supervision improves structure learning for graph neural networks." Advances in Neural Information Processing Systems 34 (2021): 22667-22681.
>
> Liu, Yixin, et al. "Towards unsupervised deep graph structure learning." Proceedings of the ACM Web Conference 2022. 2022.
>
> _Regarding Comment 2 on Theorem 3.1._ We agree that the phrase "$W$ does not depend on $A$" under the joint optimization setting is not clear enough, in particular the word "depend".
> We will clarify this phrase in the final version. We do that by stating that we address $\min_{A,W} F$ using gradient descent iterates to simultaneously update $A$ and $W$ at each iteration. Therefore by construction, $\frac{\partial W_t}{\partial A_t} = 0$ at any iteration $t$, and this derivative doesn't appear when computing the gradient for iteration $t+1$. This is what we mean by saying that at each iteration $W$ doesn't depend on $A$ through joint optimization updates.
>
> _Regarding Comment 3 on Theorem 3.1._ This is an interesting point.
> Indeed, our theory for the bilevel setting addresses the alternating optimization setting.
> Recall that we do one outer iteration after $\tau_{in}$ inner iterations, and so on.
> Lets now imagine that instead of this one outer iteration, we do $\tau_{in}$ consecutive outer iterations after $\tau_{in}$ inner iterations, and so on. This is alternating optimization with $\tau_{in}$ iterations in every round.
> Theorem 3.3 can be extended to show that hypergradient scarcity emerges here too, because its requirements are satisfied for every one of the consecutive outer iterations. For completeness, this requirement is that the inner parameters $W_{\tau_{in}}$ don't depend on affected edges.
>
> _k-hop vs k-layer._ Indeed, this might be confusing to the reader. The answer to your question is yes: the number of the GNN layers defines the number of hops not affected by hypergradient scarcity.
> To mitigate any confusion, we will emphasize this fact and point out the link of the number of layers and the number of hops, which both are $k$.
>
> _Regarding the chain rule._ We will take this comment into account in the final version.
>
> _Regarding the experiments._ We will point out the purpose of having the synthetic dataset 1, which is a regression problem and validating our findings on regression problems, particularly with the Laplacian regularization in the inner problem.
> Regarding the impact of the distribution of labelled nodes on the decrease rate of hypergradient magnitude, this is a very interesting observation that we couldn't justify _formally_ and is left for a future work, which we state in the manuscrit.
>
> _Regarding the distance._ We agree that a formal definition of each distance used in the document would be better. We will incorporate such definitions in the final version of the paper.

---

### Review · Reviewer_TBy1 · 2024-05-04

**Summary Of Contributions:**

The paper investigates the issue of gradient scarcity in graph learning under semi-supervised conditions, where edges between distant unlabeled nodes receive zero gradients. This phenomenon, initially identified in shallow Graph Neural Networks (GNNs) using a single loss function, is mathematically characterized and shown to also occur in bilevel optimization. Despite infinite receptive fields in models like those using Laplacian regularization, gradient scarcity persists due to exponential decreases in gradient amplitude with distance from labeled nodes. Various solutions are explored, including latent graph learning and graph regularization. Empirical results validate these findings and indicate that gradient scarcity also affects models like Approximate Personalized Propagation of Neural Predictions (APPNP), suggesting its prevalence across different graph learning models.

**Audience:**

Yes

**Claims And Evidence:**

Yes

**Requested Changes:**

Please refer to the Weanknesses part.

**Strengths And Weaknesses:**

Strengths:

1. The paper provides comprehensive background information and clear illustrations, enabling readers unfamiliar with gradient scarcity in graph learning to quickly grasp the concepts discussed.
2. The paper's approach is both natural and principled, supported by extensive theoretical analysis.
3. The results pertaining to the Approximate Personalized Propagation of Neural Predictions (APPNP) appear promising.

Weaknesses:
The paper is fundamentally solid, yet I recommend some revisions to enhance its structure and clarity.

1. The introduction is formally correct in defining the tasks, but incorporating natural language and visual aids could make the paper more engaging.
2. The section on related work contains excessive content on GNNs and graph learning, much of which is general and appears redundant, particularly in contrast to the short introduction of bilevel optimization.
3. In the tables, employing color coding or bold text could help emphasize key results.

---

> ### Author Response · Authors · 2024-05-15
>
> We would like to thank the reviewer for the detailed feedback and the constructive comments.
> We are glad that the reviewer found the paper fundamentally solid.
>
> _Regarding the introduction._ In the final version, we will clarify and explain the key concepts and equations in a natural manner that is more reader-friendly.
>
> _Regarding the length of literature review._ Indeed, the related work section is excessive on GNNs and graph learning. We will shorten it in the final version and make it more concise.
>
> _Regarding the typography._ Indeed, this modification makes results easier to read and follow.
> We will consider employing the bold text in the final version.

---

### Review · Reviewer_2MfE · 2024-05-11

**Summary Of Contributions:**

The authors demonstrate the phenomenon of gradient scarcity in the bilevel optimization setting where separate loss functions are used for each of the graphs and the model. They show that gradient scarcity occurs not just because of finite receptive fields, but also in Laplacian regularization. They investigate several solutions and produce empirical results to validate their theoretical analysis.

**Audience:**

Yes

**Broader Impact Concerns:**

N/A.

**Claims And Evidence:**

Yes

**Requested Changes:**

The main potential small issue is that the authors choose to study a specific problem in a very specific setting. If would be great if the authors could add a paragraph or two expanding on and giving context to the importance of the bilevel-optimization setting in comparison to other settings.

I have no other requested changes. I recommend acceptance.

**Strengths And Weaknesses:**

Strengths:

- Clarity: the paper is presented in a clear way and the material is motivated well
- Experiments: The experiments presented are rather comprehensive and they match well in terms of evaluating and supporting the claims made in the theoretical part of the paper
- Theory: the theoretical portion of the paper is 1. presented clearly 2. to the best of my knowledge, correct 3. sufficiently deep and relevant.

To the best of my knowledge, the results presented are sound, and the topic is of relevance to the audience of TMLR.

- There are no real obvious weakness in the paper...all the bases are covered pretty well.

---

> ### Author Response · Authors · 2024-05-15
>
> We thank the reviewer for the detailed feedback and the constructive comments.
> We are glad that the reviewer found the paper clear and well-written.
>
> _Regarding the setting_.
> The use of bilevel optimization for graph learning is recent.
> To the best of our knowledge, there are two works that propose bilevel learning of graphs in the literature, these are Franceschi et al. (2019) and Wan and Kokel (2021).
> Following your comment, we will highlight two main differences between the bilevel and the joint frameworks in the final version.
> The first difference is the way the graph is perceived.
> Indeed, the bilevel framework perceives it as a hyper-parameter, whereas the joint one perceives it in a similar way to GNN weights.
> This is expected, as the stream of bilevel graph learning follows the advances in bilevel programming for hyperparameter optimization and meta learning.
> The second difference is that the joint framework is more prone to overfitting, which is mitigated in the literature by graph regularization and similar techniques.
> Conducting further comparisons between both frameworks in graph learning in terms of performance and limitations is indeed a very interesting matter that we will consider in our future works.
>
> Franceschi, Luca, et al. "Learning discrete structures for graph neural networks." International conference on machine learning. PMLR, 2019.
>
> Wan, Guihong, and Harsha Kokel. "Graph sparsification via meta-learning." DLG@ AAAI (2021).

---

### Author Response · Authors · 2024-05-15
**General comment on the reviews**

We are grateful for the detailed and constructive feedback from all reviewers.
We are glad that all referees seems to be very positive with our submission and we are happy to address the comments and suggestions for each referee individually.
We would like to highlight that the referees consider our submission as _fundamentally solid_, _presented in a clear way_, _motivated well_, _sufficiently deep and relevant_, contains _comprehensive background information_ and _ambitious and with many very interesting contributions_ to mention a few.

The modification are highlighted in red in the revision submitted.

---

> ### Comment · Reviewer_2MfE · 2024-05-28
> **Official Comment**
>
> The authors responses sufficiently addressed my comments. I recommend acceptance, provided that the authors follow through on the updates/citations that they will add to their paper as detailed in their reply.

---

### Comment · Editors_In_Chief · 2025-12-02

Congratulations to the authors on this paper being named a 2025 Outstanding Certification Finalist!

For more information, see https://medium.com/@TmlrOrg/announcing-the-2025-tmlr-outstanding-certification-e26d548ff011.

---

### Decision · Action_Editor_4cHG · 2024-06-11

**Recommendation:** Accept as is

**Comment:**

This paper study the problem of gradient scarcity (zero gradients) that appears when learning graphs with objective value only on a subset of the nodes. The paper show that this problem occurs also with bilevel optimization and that gradients remains small with an exponential attenuation even when using Laplacian regularization. The authors the discuss different approaches to mitigate the problem and illustrate their findings in numerical experiments.

All reviewers found the paper well written and of high quality with significant contributions with theoretical results and numerical experiments in support of those results. The few comments and requests for clarification were all answered in the reply and corresponding modifications were done in the submission. All reviewers recommended an acceptance during the discussion.

Due to the quality of the paper that brings novel understanding on the way information is propagated (or not) in graph learning the AE recommends a feature certification.

**Audience:**

The paper is of definite interest to the GNN and graph learning community.

**Claims And Evidence:**

All the claims in the paper are supported bay theoretical proofs and numerical simulations.